# Exploiting Hidden Structures in Non-Convex Games for Convergence to Nash Equilibrium

**Iosif Sakos**[*]
Engineering Systems and Design (SUTD)
iosif_sakos@mymail.sutd.edu.sg

**Emmanouil V. Vlatakis-Gkaragkounis**[*]
UC Berkeley
emvlatakis@cs.columbia.edu

**Panayotis Mertikopoulos**
Univ. Grenoble Alpes, CNRS, Inria, Grenoble INP
LIG 38000 Grenoble, France
panayotis.mertikopoulos@imag.fr

**Georgios Piliouras**
Engineering Systems and Design (SUTD)
georgios@sutd.edu.sg

## Abstract

A wide array of modern machine learning applications – from adversarial models to multi-agent reinforcement learning – can be formulated as non-cooperative games whose Nash equilibria represent the system's desired operational states. Despite having a highly non-convex loss landscape, many cases of interest possess a latent convex structure that could potentially be leveraged to yield convergence to an equilibrium. Driven by this observation, our paper proposes a flexible first-order method that successfully exploits such "hidden structures" and achieves convergence under minimal assumptions for the transformation connecting the players' control variables to the game's latent, convex-structured layer. The proposed method – which we call preconditioned hidden gradient descent (PHGD) – hinges on a judiciously chosen gradient preconditioning scheme related to natural gradient methods. Importantly, we make no separability assumptions for the game's hidden structure, and we provide explicit convergence rate guarantees for both deterministic and stochastic environments.

## 1 Introduction

Many powerful AI architectures are based on the idea of combining conceptually straightforward settings coming from game theory with the expressive power of neural nets. Some prominent examples of this type include generative adversarial networks (GANs) [14], robust reinforcement learning [36], adversarial training [26], multi-agent reinforcement learning in games [35, 39, 41], and even multi-player games that include free-form natural-language communication [4]. Intuitively, in all these cases, the game-theoretic abstraction serves to provide a palpable, easy-to-understand target, i.e., an equilibrium solution with strong axiomatic justification. However, from a complexity-theoretic standpoint, such targets are excessively ambitious, requiring huge amounts of data to express and compute, even approximately. Because of this, the agents' policies must be encoded via a universal function approximator (such as a neural net) and training this architecture boils down to iteratively updating these parameters until the process – hopefully! – converges to the target equilibrium.

Unfortunately, despite the ubiquitousness of these settings, the design of algorithms with provable convergence guarantees is still relatively lacking. This deficit is not surprising if one considers that even the – comparatively much simpler – problem of equilibrium learning in finite games is hindered by numerous computational hardness [9, 10] as well as dynamic impossibility results

---

[*]Equal Contribution.

37th Conference on Neural Information Processing Systems (NeurIPS 2023).

[15, 16, 19, 24, 30, 31]. In this regard, our best hope for designing provably convergent algorithms is to focus on specific classes of games with some *useful structure* to exploit.

One of the most well-established frameworks of this type is the class of *monotone games* whose study goes back at least to Rosen [37]. As special cases, this setting includes single-agent convex minimization problems, two-player convex-concave min-max games, diagonally convex $N$-player games, etc. Owing to this connection, there has been a proliferation of strong positive results at the interface of game theory and optimization, see e.g., [28, 38] and references therein. In our case however, the agents do not play this monotone game directly, but can only access it *indirectly* via an encoding layer of *control variables* – like the weight parameters of a neural net that outputs a feasible strategy profile for the game in question. In this sense, from a machine learning perspective, the strategies of the game are *latent variables*, so we can think of each player as being equipped with a smooth mapping from a high-dimensional space of control variables to the strategy space of the game. Importantly, in contrast to the control variables, the latent variables are not directly accessible to the players themselves and should only be viewed as auxiliary variables – to all extents and purposes, the goal remains to find an operationally desirable control layer configuration. In this sense, the convex structure of the game becomes "hidden" behind the control layer, which entangles multiple input/control variables into nonlinear manifolds of latent variables.

As a result, the convex structure of the underlying game is effectively destroyed, resulting in highly non-convex end-to-end interactions. This raises the following central challenge:

> *Can we design provably convergent algorithms for non-convex games with a hidden structure in the presence of general couplings between control and latent variables?*

Prior work in the area has shown that this is a promising and, at the same time, highly challenging question. In [42], the setting of hidden bilinear games was introduced and a number of negative results were presented, to the effect that gradient-descent-ascent can exhibit a variety of non-convergent behaviors, even when the game admits a hidden *bilinear* structure. Subsequently, [13] provided an approximate minimax theorem for a class of two-agent games where the players pick neural networks, but did not provide any convergent training algorithm for this class of games. Instead, the first positive result on the dynamics of hidden games was obtained by [43] who established a series of non-local convergence guarantees to the von Neumann max-min solution of the game in the case of two-agent hidden strictly convex-concave games for all initial conditions satisfying a certain genericity assumption. This approach, however, only applied to *continuous-time dynamics – not algorithms –* and it further imposed strong separability assumptions on the representation of the game in the control layer. More recently, [32] established the first global convergence guarantees in hidden games but, once again, these apply only to *continuous-time dynamics* and a special case of two-agent convex-concave games (akin to playing a convex combination of hidden games with one-dimensional latent spaces). This paper is the closest antecedent to our work as it introduces a dynamical system, called Generalized Natural Gradient Flow, which makes the $L^2$ norm between the equilibrium in the hidden game and the current set of latent variables a Lyapunov function for the system.

**Our results & techniques.** Our paper seeks to provide an affirmative answer to the key challenge above under minimal assumptions on the coupling between latent and control variables. To that effect, we only assume that each agent is able to affect a measurable change along any latent variable by updating their control variables appropriately; without this assumption spurious equilibria can emerge due to the deficiency of the control layer architecture. Importantly however, even though the map from control to latent variables is known to the players, *we do not assume* that it can be efficiently inverted (e.g., to solve for a profile of control variables that realizes a profile of latent variables). Otherwise, if it could, the entire game could be solved directly in the latent layer and then ported back to the control layer, thus rendering the whole problem moot – and, indeed, when working with realistic neural net architectures, this inversion problem is, to all intents and purposes, impossible.

For intuition, we begin by designing a new continuous-time flow, that we call preconditioned hidden gradient dynamics, and which enjoys strong convergence properties in games with a hidden strictly monotone structure (Proposition 1). Similarly to [32], this is achieved by using the $L^2$ norm in the latent layer as a Lyapunov function in the control layer; however, the similarities with the existing literature end there. Our paper does not make any separability or low-dimensionality assumptions, and is otherwise *purely algorithmic:* specifically, building on the continuous-time intuition, we provide a concrete, implementable algorithm, that we call *preconditioned hidden gradient descent* (PHGD);

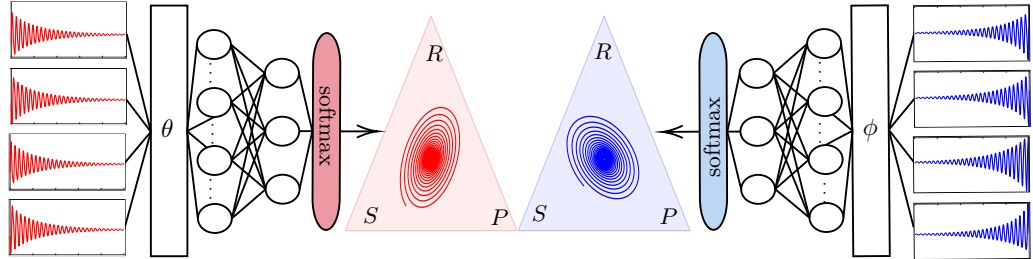

**Figure 1:** A hidden game of Rock-Paper-Scissors with strategies encoded by two multi-layer perceptrons (MLPs), whose 4-dimensional input is dictated by the two players (cf. [32]). The nonlinearity of the MLP representation maps leads to a highly non-convex non-concave zero-sum game. However, by employing PHGD, both players' MLP control variables accurately identify the $(1/3, 1/3, 1/3)$ equilibrium in the game's latent space.

this algorithm is run with *stochastic gradients*, and enjoys a series of strong, global convergence guarantees in hidden games.

First, as a baseline, Theorem 1 shows that a certain averaged process achieves an $\mathcal{O}(1/\sqrt{t})$ convergence rate in all games with a hidden monotone structure and Lipschitz continuous loss functions. If the hidden structure is strongly monotone, Theorem 2 further shows that this rate can be improved to $\mathcal{O}(1/t)$ for the *actual* trajectory of the players' control variables; and if the algorithm is run with full, deterministic gradients, the rate becomes *geometric* (Theorem 3). To the best of our knowledge, these are the first bona fide algorithmic convergence guarantees for games with a hidden structure.

## 2 Problem setup and preliminaries

Throughout the sequel, we will focus on continuous $N$-player games where each player, indexed by $i \in \mathcal{N} \coloneqq \{1, \ldots, N\}$, has a convex set of *control variables* $\theta_i \in \Theta_i \coloneqq \mathbb{R}^{m_i}$, and a continuously differentiable *loss function* $\ell_i \colon \Theta \to \mathbb{R}$, where $\Theta \coloneqq \prod_i \Theta_i$ denotes the game's *control space*. For concreteness, we will refer to the tuple $\Gamma \equiv \Gamma(\mathcal{N}, \Theta, \ell)$ as the *base game*.

The most relevant solution concept in this setting is that of a *Nash equilibrium*, i.e., an action profile $\theta^* \in \Theta$ that discourages unilateral deviations. Formally, we say that $\theta^* \in \Theta$ is a *Nash equilibrium* of the base game $\Gamma$ if

$$\ell_i(\theta^*) \leq \ell_i(\theta_i; \theta^*_{-i}) \quad \text{for all } \theta_i \in \Theta_i, i \in \mathcal{N} \tag{NE}$$

where we employ the standard game-theoretic shorthand $(\theta_i; \theta_{-i})$ to distinguish between the action of the $i$-th player and that of all other players in the game. Unfortunately, designing a learning algorithm that provably outputs a Nash equilibrium is a very elusive task: the impossibility results of Hart & Mas-Colell [15, 16] already preclude the existence of uncoupled dynamics that converge to a Nash equilibrium in all games; more recently, Milionis et al. [31] established a similar impossibility result even for possibly *coupled* dynamics (in both discrete and continuous time), while Daskalakis et al. [9, 10] has shown that even the *computation* of an approximate equilibrium can be beyond reach.

In view of this, our work focuses on games with a hidden, *latent* structure that can be exploited to compute its Nash equilibria. More precisely, inspired by [42, 43] we have the following definition.

**Definition 1.** We say that the game $\Gamma \equiv \Gamma(\mathcal{N}, \Theta, \ell)$ admits a *latent – or hidden – structure* if:

1. Each player's control variables can be mapped faithfully to a closed convex set of *latent variables* $x_i \in \mathcal{X}_i \subseteq \mathbb{R}^{d_i}$; formally, we posit that there exists a Lipschitz smooth map $\chi_i \colon \Theta_i \to \mathcal{X}_i$ with no critical points and such that $\mathrm{cl}(\chi_i(\Theta_i)) = \mathcal{X}_i$.

2. Each player's loss function factors through the game's *latent space* $\mathcal{X} \coloneqq \prod_i \mathcal{X}_i$ as

$$\ell_i(\theta) = f_i(\chi_1(\theta_1), \ldots, \chi_N(\theta_N)) \tag{1}$$

for some Lipschitz smooth function $f_i \colon \mathcal{X} \to \mathbb{R}$ called the player's *latent loss function*.

For concreteness, we will refer to the product map $\chi(\theta) = (\chi_i(\theta_i))_{i \in \mathcal{N}}$ as the game's *representation map*, and the tuple $\mathcal{G} \equiv \mathcal{G}(\mathcal{N}, \mathcal{X}, f)$ will be called the *hidden/latent game*. To simplify notation later on, we also assume that $\mathcal{X}_i$ has nonempty topological interior, so, in particular, $\dim(\mathcal{X}_i) = d_i \leq m_i$.

We illustrate the above notions in two simple – but not simplistic – examples below; for a schematic representation, cf. Fig. 2.

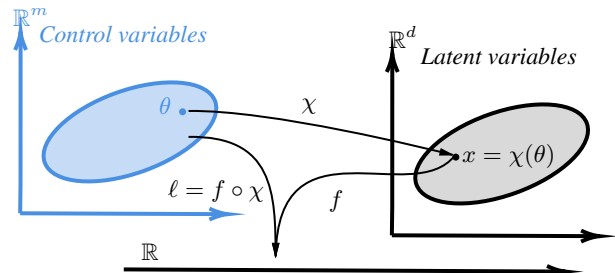

**Figure 2:** Schematic representation of a game with a hidden / latent structure.

**Example 2.1.** Consider a single-agent game ($N = 1$) where the objective is to minimize the non-convex function $\ell(\theta) = \sum_{\alpha=1}^{|\mathcal{D}|} (\mathrm{sigmoid}(\theta_\alpha) - s_\alpha)^2$ over a dataset $\mathcal{D} = \{s_\alpha\}$. This problem can be recast as a hidden convex problem with $f(x) = \sum_{\alpha=1}^{|\mathcal{D}|} (x_\alpha - s_\alpha)^2$ and $\chi(\theta) = \mathrm{sigmoid}(\theta)$.

**Example 2.2.** A more complex scenario involves non-convex / non-concave min-max optimization problems of the form $\min_{\theta_1} \max_{\theta_2} \chi_1(\theta_1)^\mathsf{T} C \chi_2(\theta_2)$ where $\chi_1$ and $\chi_2$ are preconfigured multi-layer perceptrons (MLPs) constructed with smooth activation functions (such as CeLUs). This problem can be reformulated as a hidden bilinear problem by setting $f_1(x_1, x_2) = x_1^\mathsf{T} C x_2 = -f_2(x_1, x_2)$.

Before moving forward, there are some points worth noting regarding the above definitions.

*Remark* 1. First, the requirement $\mathrm{cl}(\chi(\Theta)) = \mathcal{X}$ means that all latent variable profiles $x \in \mathcal{X}$ can be approximated to arbitrary accuracy in the game's control space. The reason that we do not make the stronger assumption $\chi(\Theta) = \mathcal{X}$ is to capture cases like the sigmoid map: if $\chi(\theta) = [1 + \exp(-\theta)]^{-1}$ for $\theta \in \mathbb{R}$, the image of $\chi$ is the interval $(0, 1)$, which is convex but not closed.

*Remark* 2. Second, the requirement that $\chi$ has no critical points simply means that, for any control variable configuration $\theta \in \Theta$, the Jacobian $\mathrm{Jac}(\chi(\theta))$ of $\chi$ at $\theta$ has full rank. By the implicit function theorem [23], this simply means that $\chi$ locally looks like a projection (in suitable coordinates around $\theta$), so it is possible to affect a measurable change along any feasible latent direction by updating each player's control variables appropriately. This is no longer true if $\chi$ has critical points, so this is a minimal requirement to ensure that no spurious equilibria appear in the game's latent space.

To proceed, we will assume that the latent game is *diagonally convex* in the sense of Rosen [37], a condition more commonly known in the optimization literature as *monotonicity* [12]. Formally, let

$$g_i(x) = \nabla_i f_i(x) \coloneqq \nabla_{x_i} f_i(x) \tag{2}$$

denote the individual gradient of the latent loss function of player $i \in \mathcal{N}$, and write $g(x) = (g_1(x), \ldots, g_N(x))$ for the profile thereof. We then say that the latent game $\mathcal{G}$ is:

- *Monotone* if $\langle g(x') - g(x), x' - x \rangle \geq 0$ for all $x, x' \in \mathcal{X}$. (3a)
- *Strictly monotone* if (3a) holds as a strict inequality whenever $x' \neq x$. (3b)
- *Strongly monotone* if $\langle g(x') - g(x), x' - x \rangle \geq \mu \|x' - x\|^2$ for some $\mu > 0$ and all $x, x'$. (3c)

Clearly, strong monotonicity implies strict monotonicity, which in turn implies monotonicity; on the other hand, when we want to distinguish between problems that are monotone but not strictly monotone, we will say that $g$ is *merely monotone*. Finally, extending the above to the base game, we will say that $\Gamma$ admits a *hidden monotone structure* when the latent game $\mathcal{G}$ is monotone as above (and likewise for strictly / strongly monotone structures).

Examples of monotone games include neural Kelly auctions and Cournot oligopolies [5, 6, 20, 25], covariance matrix optimization problems and power control [27, 29, 38], certain classes of congestion games [34], etc. In particular, in the case of convex minimization problems, monotonicity (resp. strict / strong monotonicity) corresponds to convexity (resp. strict / strong convexity) of the problem's objective function. In all cases, it is straightforward to check that the image $x^* = \chi(\theta^*)$ of a Nash equilibrium $\theta^* \in \Theta$ of the base game satisfies a Stampacchia variational inequality of the form

$$\langle g(x^*), x - x^* \rangle \geq 0 \quad \text{for all } x \in \mathcal{X}. \tag{SVI}$$

In turn, by monotonicity, this characterization is equivalent to the Minty variational inequality

$$\langle g(x), x - x^* \rangle \geq 0 \quad \text{for all } x \in \mathcal{X}. \tag{MVI}$$

To avoid trivialities, we will assume throughout that the solution set $\mathcal{X}^*$ of (SVI)/(MVI) is nonempty. This is a standard assumption without which the problem is not well-posed – and hence, meaningless from an algorithmic perspective.

**Notation.** To streamline notation (and unless explicitly mentioned otherwise), we will denote control variables by $\theta_i$ and $\theta$, and we will write $x_i = \chi_i(\theta_i)$ and $x = \chi(\theta)$ for the induced latent variables. We will also write $m := \sum_i m_i$ for the dimensionality of the game's control space $\Theta$ and $d := \sum_i d_i$ for the dimensionality of the latent space $\mathcal{X}$ (so, in general, $m \geq d$). Finally, when the representation map $\chi$ is clear from the context, we will write $\mathbf{J}_i(\theta_i) := \mathrm{Jac}(\chi_i(\theta_i)) \in \mathbb{R}^{d_i \times m_i}$ for the Jacobian matrix of $\chi_i$ at $\theta_i \in \Theta_i$, and $\mathbf{J}(\theta) = \bigoplus_i \mathbf{J}_i(\theta_i)$ for the associated block diagonal sum.

# 3 Hidden gradients and preconditioning

We are now in a position to present our main algorithmic scheme for equilibrium learning in games with a hidden monotone structure. In this regard, our aim will be to overcome the following limitations in the existing literature on learning in hidden games: (*i*) the literature so far has focused exclusively on continuous-time dynamics, with no discrete-time algorithms proven to efficiently converge to a solution; (*ii*) the number of players is typically limited to two; and (*iii*) the representation maps are *separable* in the sense that the control variables are partitioned into subsets and each subset affects exactly one latent variable.[1]

We start by addressing the last two challenges first. Specifically, we begin by introducing a gradient preconditioning scheme that allows us to design a convergent *continuous-time* dynamical system for hidden monotone games with an *arbitrary* number of players and *no separability* restrictions for its representation maps. Subsequently, we propose a bona fide algorithmic scheme – which we call *preconditioned hidden gradient descent* (PHGD) – by discretizing the said dynamics, and we analyze the algorithm's long-run behavior in Section 4.

**3.1. Continuous-time dynamics for hidden games.** To connect our approach with previous works in the literature, our starting point will be a simple setting already captured within the model of [32], min-max games with a hidden convex-concave structure and unconstrained one-dimensional control and latent spaces per player, i.e., $\Theta_i = \mathbb{R} = \mathcal{X}_i$ for $i = 1, 2$. We should of course note that this specific setting is fairly restrictive and not commonly found in deep neural network practice: in real-world deep learning applications, neural nets generally comprise multiple interconnected layers with distinct activation functions, so the input-output relations are markedly more complex and intertwined. Nevertheless, the setting's simplicity makes the connection with natural gradient methods particularly clear, so as in [32], it will serve as an excellent starting point.

Concretely, Mladenovic et al. [32] analyze the *natural hidden gradient dynamics*

$$\dot{\theta}_i = -\frac{1}{|\chi_i'(\theta_i)|^2} \frac{\partial \ell_i}{\partial \theta_i} \quad \text{for all } i \in \mathcal{N}. \tag{NHGD}$$

The reason for this terminology – and the driving force behind the definition of (NHGD) – is the observation that, in one-dimensional settings, a direct application of the chain rule to the defining relation $\ell_i = f_i \circ \chi$ of the game's latent loss functions yields $\partial \ell_i / \partial \theta_i = \chi_i'(\theta_i) \cdot \partial f_i / \partial x_i$ so, in turn, (NHGD) becomes

$$\dot{\theta}_i = -\frac{1}{\chi_i'(\theta_i)} \frac{\partial f_i}{\partial x_i} \tag{4}$$

where, for simplicity, we are tacitly assuming that $\chi_i(\theta_i)' > 0$.

This expression brings two important points to light. First, even though (NHGD) is defined in terms of the actual, control-layer gradients $\partial \ell_i / \partial \theta_i$ of $\Gamma$, Eq. (4) shows that the dynamics are actually driven by the latent-layer, "hidden gradients" $g_i(x) = \partial f_i / \partial x_i$ of $\mathcal{G}$. Second, the preconditioner $|\chi_i'(\theta_i)|^{-2}$ in (NHGD) can be seen as a Riemannian metric on $\Theta$: instead of defining gradients relative to the standard Euclidean metric of $\Theta \equiv \mathbb{R}^N$, (NHGD) can be seen as a gradient flow relative

---

[1]The separability assumption also rules out the logit representation $\exp(\theta_{i\alpha}) \big/ \sum_\beta \exp(\theta_{i\beta})$ that is standard when the latent structure expresses mixed strategies in a finite game.

to the Riemannian metric $g_{ij}(\theta) = \delta_{ij}\chi'_i(\theta_i)$, which captures the "natural" geometry induced by the representation map $\chi$.[2]

From an operational standpoint, the key property of (NHGD) that enabled the analysis of [32] is the observation that the $L^2$ energy function

$$E(\theta) = \tfrac{1}{2}\|\chi(\theta) - x^*\|^2 \tag{5}$$

between the latent representation $x = \chi(\theta)$ of $\theta$ and a solution $x^*$ of (SVI) / (MVI) is a Lyapunov function for (NHGD). Indeed, a straightforward differentiation yields

$$\dot{E}(\theta) = \frac{1}{2}\frac{d}{dt}\|x - x^*\|^2 = \sum_i \dot{x}_i \cdot (x_i - x_i^*) = -[g(\chi(\theta))]^\intercal(\chi(\theta) - x^*) \leq 0 \tag{6}$$

with the penultimate step following from (4) and the last one from (MVI). It is then immediate to see that the latent orbits $x(t) = \chi(\theta(t))$ of (NHGD) converge to equilibrium in strictly monotone games.

The approach in the example above depends crucially on the separability assumption which, among others, trivializes the problem. Indeed, if there is no coupling between control variables in the game's latent space, the equations $x_i = \chi_i(\theta_i)$ can be backsolved easily for $\theta$ (e.g., via binary search), so it is possible to move back-and-forth between the latent and control layers, ultimately solving the game in the latent layer and subsequently extracting a solution configuration in the control layer. Unfortunately however, extending the construction of (NHGD) to a non-separable setting is not clear, so it is likewise unclear how to exploit the hidden structure of the game beyond the separable case.

To that end, our point of departure is the observation that the Lyapunov property (6) of the energy function $E(\theta)$ is precisely the key feature that enables convergence of (NHGD). Thus, assuming that control variables are mapped to latent ones via a general – but possibly highly coupled – representation map $\chi\colon \Theta \to \mathcal{X}$, we will consider an abstract preconditioning scheme of the form

$$\dot{\theta}_i = -\mathbf{P}_i(\theta_i)v_i(\theta) \tag{7}$$

where

$$v_i(\theta) \coloneqq \nabla_{\theta_i}\ell_i(\theta) \tag{8}$$

denotes the individual loss gradient of player $i$, while the preconditioning matrix $\mathbf{P}_i(\theta_i) \in \mathbb{R}^{m_i \times m_i}$ is to be designed so that (6) still holds under (7). In this regard, a straightforward calculation (which we prove in Appendix A) yields the following:

**Lemma 1.** *Under the dynamics* (7)*, we have*

$$\dot{E}(\theta) = -\sum\nolimits_{i \in \mathcal{N}}[g_i(\chi(\theta))]^\intercal\mathbf{J}_i(\theta_i)\mathbf{P}_i(\theta_i)[\mathbf{J}_i(\theta_i)]^\intercal(\chi_i(\theta_i) - x_i^*) \tag{9}$$

*where, as per Section 2, $\mathbf{J}_i(\theta_i) \in \mathbb{R}^{d_i \times m_i}$ denotes the Jacobian matrix of the map $\chi_i\colon \Theta_i \to \mathcal{X}_i$. More compactly, letting $\mathbf{P} \coloneqq \bigoplus_i \mathbf{P}_i$ denote the block-diagonal ensemble of the players' individual preconditioning matrices $\mathbf{P}_i$ (and suppressing control variable arguments for concision), we have*

$$\dot{E} = -g(x)^\intercal \cdot \mathbf{J}\mathbf{P}\mathbf{J}^\intercal \cdot (x - x^*) \tag{10}$$

In view of Lemma 1, a direct way to achieve the target Lyapunov property (6) would be to find a preconditioning matrix $\mathbf{P}$ such that $\mathbf{J}\mathbf{P}\mathbf{J}^\intercal = \mathbf{I}$. However, since $\mathbf{J}$ is surjective (by the faithfulness assumption for $\chi$), the Moore-Penrose inverse $\mathbf{J}^+$ of $\mathbf{J}$ will be a right inverse to $\mathbf{J}$, i.e., $\mathbf{J}\mathbf{J}^+ = \mathbf{I}$. Hence, letting $\mathbf{P} = (\mathbf{J}^\intercal\mathbf{J})^+ = \mathbf{J}^+[\mathbf{J}^+]^\intercal$, we obtain:

$$\mathbf{J}\mathbf{P}\mathbf{J}^\intercal = \mathbf{J}(\mathbf{J}^\intercal\mathbf{J})^+\mathbf{J}^\intercal = \mathbf{J}\mathbf{J}^+(\mathbf{J}^\intercal)^+\mathbf{J}^\intercal = \mathbf{I}\cdot\mathbf{I} = \mathbf{I} \tag{11}$$

In this way, unraveling the above, we obtain the *preconditioned hidden gradient flow*

$$\dot{\theta}_i = -\mathbf{P}_i(\theta_i)\nabla_i\ell_i(\theta_i) \quad \text{with} \quad \mathbf{P}_i(\theta_i) = [\mathbf{J}_i(\theta_i)^\intercal\mathbf{J}_i(\theta_i)]^+ \tag{PHGF}$$

By virtue of design, the following property of (PHGF) is then immediate:

**Proposition 1.** *Suppose that $\Gamma$ admits a latent strictly monotone structure in the sense of* (3b)*. Then the energy function* (5) *is a strict Lyapunov function for* (PHGF)*, and every limit point $\theta^*$ of $\theta(t)$ is a Nash equilibrium of $\Gamma$.*

---

[2]We do not attempt to provide here a primer on Riemannian geometry; for a masterful introduction, see [22].

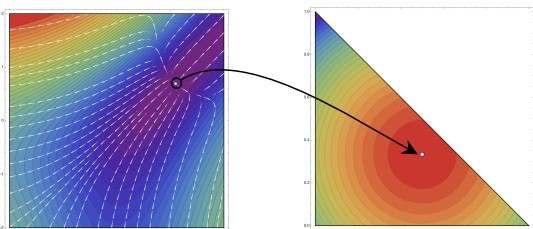

**Figure 3:** Exploiting a hidden convex structure in the control layer. On the left, we present the dynamics (PHGF) in the example of minimizing the simple function $f(\theta) = \mathrm{KL}(\mathrm{logit}(\theta_1, \theta_2) \| (1/2, 1/3, 1/6))$ over $\Theta \equiv \mathbb{R}^2$. In the subfigure to the right, we illustrate the hidden convex structure of the energy landscape, from the non-convex sublevel sets of $\Theta$ to the latent space $\mathcal{X} \equiv \{(x_1, x_2) \in \mathbb{R}^2_{\geq 0} : x_1 + x_2 \leq 1\}$.

Proposition 1 validates our design choices for the preconditioning matrix $\mathbf{P}$ as it illustrates that the dynamics (PHGF) converge to Nash equilibrium, without any separability requirements or dimensionality restrictions; in this regard, Proposition 1 already provides a marked improvement over the corresponding convergence result of [32] for (NHGD). To streamline our presentation, we defer the proof of Lemma 1 and Proposition 1 to Appendix A, and instead proceed directly to provide a bona fide, algorithmic implementation of (PHGF).

**3.2. Preconditioned hidden gradient descent.** In realistic machine learning problems, any algorithmic scheme based on (PHGF) will have to be run in an iterative, discrete-time environment; moreover, due to the challenges posed by applications with large datasets and optimization objectives driven by the minimization of empirical risk, we will need to assume that players only have access to a stochastic version of their full, deterministic gradients. As such, we will make the following blanket assumptions that are standard in the stochastic optimization literature [17, 21]:

**Assumption 1.** Each agent's loss function is an expectation of a random function $L_i \colon \Theta \times \Omega \to \mathbb{R}$ over a complete probability space $(\Omega, \mathcal{F}, \mathbb{P})$, i.e., $\ell_i(\theta) = \mathbb{E}[L_i(\theta; \omega)]$. We further assume that:

1. $L_i(\theta; \omega)$ is measurable in $\omega$ and $\beta$-Lipschitz smooth in $\theta$ (for all $\theta \in \Theta$ and $\omega \in \Omega$ respectively).
2. The gradients of $L_i$ have bounded second moments, i.e., $\sup_{\theta \in \Theta} \mathbb{E}[\|\nabla L_i(\theta; \omega)\|^2] \leq M^2$.

Taken together, the two components of Assumption 1 imply that each $\ell_i$ is differentiable and Lipschitz continuous (indeed, by Jensen's inequality, we have $\|\mathbb{E}[\nabla L_i(\theta; \omega)]\|^2 \leq \mathbb{E}[\|\nabla L_i(\theta; \omega)\|^2] \leq M^2$). Moreover, by standard dominated convergence arguments, $\nabla L_i(\theta; \omega)$ can be seen as an unbiased estimator of $\nabla \ell_i(\theta)$, that is, $\ell_i(\theta) = \mathbb{E}[\nabla L_i(\theta; \omega)]$ for all $\theta \in \Theta$. In view of this, we will refer to the individual loss gradients $\nabla_i L_i(\theta; \omega)$ of player $i \in \mathcal{N}$ as the player's individual *stochastic gradients*.

With all this in mind, the *preconditioned hidden gradient descent* algorithm is defined as the stochastic first-order recursion

$$\theta_{i,t+1} = \theta_{i,t} - \gamma_t \mathbf{P}_{i,t} V_{i,t} \tag{PHGD}$$

where:

1. $\theta_{i,t}$ denotes the control variable configuration of player $i \in \mathcal{N}$ at each stage $t = 1, 2, \dots$
2. $\gamma_t > 0$ is a variable step-size sequence, typically of the form $\gamma_t \propto 1/t^p$ for some $p \in [0, 1]$.
3. $\mathbf{P}_{i,t} \coloneqq \mathbf{P}_i(\theta_{i,t}) = [\mathbf{J}_i(\theta_{i,t})^\mathsf{T} \mathbf{J}_i(\theta_{i,t})]^+$ is the preconditioner of player $i \in \mathcal{N}$ at the $t$-th epoch.
4. $V_{i,t} \coloneqq \nabla_i L(\theta_{i,t}; \omega_t)$ is an individual stochastic loss gradient generated at the control variable configuration $\theta_t$ by an i.i.d. sample sequence $\omega_t \in \Omega$, $t = 1, 2, \dots$

The basic recursion (PHGD) can be seen as a noisy Euler discretization of the continuous flow (PHGF), in the same way that ordinary stochastic gradient descent can be seen as a noisy discretization of gradient flows. In this way, (PHGD) is subject to the same difficulties underlying the analysis of stochastic gradient descent methods; we address these challenges in the next section.

## 4  Convergence analysis and results

In this section, we present our main results regarding the convergence of (PHGD) in hidden monotone games. Because we are interested not only on the asymptotic convergence of the method but also on its *rate* of convergence, we begin this section by defining a suitable merit function for each class of hidden monotone games under consideration; subsequently, we present our main results in

Section 4.2, where we also discuss the main technical tools enabling our analysis. To streamline our presentation, all proofs are deferred to Appendix B.

**4.1. Merit functions and convergence metrics.** In the variational context of (SVI)/(MVI), the quality of a candidate solution $\hat{x}$ is typically evaluated by means of the *restricted merit function*

$$\mathrm{Gap}_{\mathcal{C}}(\hat{x}) \coloneqq \sup_{x \in \mathcal{C}} \langle g(x),\, \hat{x} - x \rangle \tag{12}$$

where the "test domain" $\mathcal{C}$ is a relatively open subset of $\mathcal{X}$ [3]. The *raison d'être* of this definition is that, if $g$ is monotone and $x^* \in \mathcal{C}$ is a solution of (SVI)/(MVI), we have

$$\langle g(x),\, x^* - x \rangle \le \langle g(x^*),\, x^* - x \rangle \le 0 \quad \text{for all } x \in \mathcal{C}, \tag{13}$$

so the supremum in (B.6) cannot be too positive if $\hat{x}$ is an approximate solution of (SVI)/(MVI). This is encoded in the following lemma, which, among others, justifies the terminology "merit function":

**Lemma 2.** *Suppose that $g$ is monotone. If $\hat{x}$ is a solution of* (SVI)/(MVI)*, we have $\mathrm{Gap}_{\mathcal{C}}(\hat{x}) = 0$ whenever $\hat{x} \in \mathcal{C}$. Conversely, if $\mathrm{Gap}_{\mathcal{C}}(\hat{x}) = 0$ and $\mathcal{C}$ is a neighborhood of $\hat{x}$ in $\mathcal{X}$, then $\hat{x}$ is a solution of* (SVI)/(MVI)*.*

Lemma 2 extends similar statements by Auslender & Teboulle [3] and **?** ]; the precise variant that we state above can be found in [**?** ], but, for completeness, we provide a proof in Appendix B.

Now, since a latent variable profile $x^* = \chi(\theta^*)$ solves (SVI) if and only if the control variable configuration $\theta^*$ is Nash equilibrium of the base game, the quality of a candidate solution $\hat{\theta} \in \Theta$ with $\hat{x} = \chi(\hat{\theta})$ can be assessed by the induced gap function

$$\mathrm{Gap}(\hat{\theta}) \coloneqq \mathrm{Gap}_{\mathcal{X}}(\chi(\hat{\theta})) = \sup_{x \in \mathcal{X}} \langle g(x),\, \hat{x} - x \rangle. \tag{14}$$

Indeed, since $\hat{x} = \chi(\hat{\theta}) \in \mathcal{X}$, Lemma 2 shows that $\mathrm{Gap}(\hat{\theta}) \ge 0$ with equality if and only if $\hat{x}$ is a Nash equilibrium of the base game $\Gamma$. In this regard, Eq. (14) provides a valid equilibrium convergence metric for $\Gamma$, so we will use it freely in the sequel as such; for a discussion of alternative convergence metrics, we refer the reader to Appendix B.

**4.2. Convergence results.** We are now in a position to state our main results regarding the equilibrium convergence properties of (PHGD). To streamline our presentation, we present our results from coarser to finer, starting with games that admit a hidden *merely* monotone structure and *stochastic* gradient feedback, and refining our analysis progressively to games with a hidden *strongly* monotone structure and *full* gradient feedback.[3] In addition, to quantify this distortion between the game's latent and control layers, we will require a technical regularity assumption for the game's representation map $\chi \colon \Theta \to \mathcal{X}$.

**Assumption 2.** The singular values of the Jacobian $\mathbf{J}(\theta)$ of the representation map $\chi$ are bounded as

$$\sigma_{\min}^2 \le \mathrm{eig}(\mathbf{J}(\theta)\mathbf{J}(\theta)^{\mathsf{T}}) \le \sigma_{\max}^2 \tag{15}$$

for some $\sigma_{\min}, \sigma_{\max} \in (0, \infty)$ and for all $\theta \in \Theta$.

With all this in hand, we begin by studying the behavior of (PHGD) in games with a hidden monotone structure.

**Theorem 1** (PHGD in hidden monotone games)**.** *Suppose that players run* (PHGD) *in a hidden monotone game with learning rate $\gamma_t \propto 1/t^{1/2}$. Then, under Assumptions 1 and 2, the averaged process $\bar{\theta}_t \in \chi^{-1}\left(t^{-1} \sum_{s=1}^{t} x_s\right)$ enjoys the equilibrium convergence rate*

$$\mathbb{E}[\mathrm{Gap}(\bar{\theta}_t)] = \mathcal{O}(\log t / \sqrt{t}). \tag{16}$$

As far as we are aware, Theorem 1 is the first result of its kind in the hidden games literature – that is, describing the long-run behavior of a discrete-time algorithm with stochastic gradient input. At the same time, it is subject to two important limitations: the first is that the averaged state $\bar{\theta}_t$ cannot be efficiently computed for general representation maps; second, even if it could, the $\mathcal{O}(\log t / \sqrt{t})$

---

[3]As expected, convergence rates improve along the way.

convergence rate is relatively slow. The two results that follow show that both limitations can be overcome in games with a hidden *strongly* monotone structure.

In this case (SVI)/(MVI) admits a (necessarily) unique solution $x^*$ in the game's control space, so we will measure convergence in terms of the latent equilibrium distance

$$\text{Err}(\hat{\theta}) := \tfrac{1}{2}\|\chi(\hat{\theta}) - x^*\|^2. \tag{17}$$

With this in mind, we have the following convergence results:

**Theorem 2** (PHGD in hidden strongly monotone games). *Suppose that players run* (PHGD) *in a hidden $\mu$-strongly monotone game with $\gamma_t = \gamma/t$ for some $\gamma > \mu$. Then, under Assumptions 1 and 2, the induced sequence of play $\theta_t \in \Theta$, $t = 1, 2, \ldots$, enjoys the equilibrium convergence rate*

$$\mathbb{E}[\text{Err}_\Theta(\theta_t)] = \mathcal{O}(1/t). \tag{18}$$

This rate is tight, even for standard strongly monotone games. To improve it further, we will need to assume that (PHGD) is run with full, *deterministic* gradients, i.e., $V_t = g(\theta_t)$. In this case, we obtain the following refinement of Theorem 2.

**Theorem 3** (PHGD with full gradient feedback in hidden strongly monotone games). *Suppose that players run* (PHGD) *in a hidden $\mu$-strongly monotone game with full gradient feedback, and a sufficiently small learning rate $\gamma > 0$. Then, under Assumptions 1 and 2, the induced sequence of play $\theta_t \in \Theta$, $t = 1, 2, \ldots$, converges to equilibrium at a geometric rate, i.e.,*

$$\text{Err}_\Theta(\theta_t) = \mathcal{O}(\rho^t) \tag{19}$$

*for some constant $\rho \in (0, 1)$ that depends only on the primitives of $\Gamma$ and the representation map $\chi$.*

Importantly, up to logarithmic factors, the convergence rates of Theorems 1–3 mirror the corresponding rates for learning in monotone games. This take-away is particularly important as it shows that, *when it exists, a hidden convex structure can be exploited to the greatest possible degree, without any loss of speed in convergence relative to standard, non-hidden convex problems.*

The proofs of Theorems 1–3 are quite involved, so we defer the details to Appendix B. That said, to give an idea of the technical steps involved, we provide below (without proof) two lemmas that play a pivotal role in our analysis. The first one hinges on a transformation of the problem's defining vector field $v(\theta) = (\nabla_i \ell_i(\theta))_{i \in \mathcal{N}}$ which, coupled with the specific choice of preconditioner $\mathbf{P}_i(\theta_i) = [\mathbf{J}_i(\theta_i)^\intercal \mathbf{J}_i(\theta_i)]^+$ in (PHGD) allows us to effectively couple the latent and control layers of the problem in a "covariant" manner:

**Lemma 3.** *Fix some $\hat{x} \in \mathcal{X}$, and consider the energy function $E(\theta; \hat{x}) = (1/2)\|\chi(\theta) - \hat{x}\|^2$. Then, for all $\theta \in \Theta$, we have*

$$\mathbf{J}(\theta)\mathbf{P}(\theta)\nabla_\theta E(\theta; \hat{x}) = \chi(\theta) - \hat{x}. \tag{20}$$

Our second intermediate result builds on Lemma 3 and provides a "template inequality" for the energy function $E(\theta; \hat{x})$ in the spirit of [7, 11, 18].

**Lemma 4** (Template inequality). *Suppose that Assumptions 1 and 2 hold. Then, with notation as in Lemma 3, the sequence $E_t := (1/2)\|\chi(\theta_t) - \hat{x}\|^2$, $t = 1, 2, \ldots$, satisfies*

$$E_{t+1} \leq E_t - \gamma_t g(x_t)^\intercal (x_t - \hat{x}) + \gamma_t \phi_t + \gamma_t^2 \psi_t, \tag{21}$$

*where $x_t := \chi(\theta_t)$, $\phi_t := (\mathbf{J}(\theta_t)\mathbf{P}(\theta_t)V_t - g(x_t))^\intercal (x_t - \hat{x})$ and $\psi_t$ is a random error sequence with $\sup_t \mathbb{E}[\psi_t] < \infty$.*

This inequality plays a pivotal role in our analysis because it allows us to couple the restricted merit function (B.8) in the game's control space with the evolution of the algorithm's quasi-Lyapunov function in the game's latent space. We provide the relevant details and calculations in Appendix B.

## 5 Experiments

This section demonstrates our method's applicability in a couple of different and insightful setups. Technical details of those setups, as well as additional experimental results are deferred to the supplementary material. We start with a regularized version of Matching Pennies zero-sum game where the players' strategies are controlled by two individual preconfigured differentiable MLPs. Each MLP acts as the player's representation map $\chi_i$, which for each input $\theta_i$ outputs a uni-dimensional latent variable $x_i = \chi_i(\theta_i)$ guaranteed to lie in $\mathcal{X}_i \equiv [0, 1]$; the player's latent space.

Figure 4 illustrates the trajectory of (PHGD), represented by the black curve. The algorithm employs a constant step-size of $0.01$ and is initialized at the arbitrary state $(1.25, 2.25)$ in the control variables' space. The color map in the figure serves as a visual representation of the level sets associated with the proposed energy function (5). Notably, the trajectory of the algorithm intersects each of the energy function's level sets at most once, indicating its non-cycling behavior, which is an issue that shows up often in the equilibration task. Due to the design of our hidden maps, the stabilization at the point $(0, 0)$ corresponds to the $(\chi_1(0), \chi_2(0)) = (\frac{1}{2}, \frac{1}{2})$ the unique equilibrium of the game.

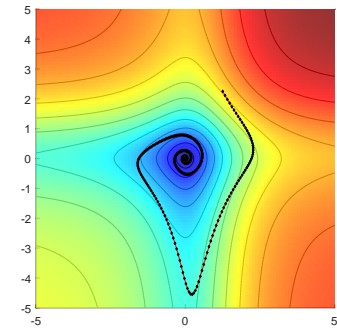

**Figure 4:** The trajectory of (PHGD) in a hidden Matching Pennies game over the sub-level sets of the energy function in (5)

In the second, more complex, example, we consider a strongly-monotone regularized modification of an (atomic) El Farol Bar congestion game among $N = 30$ players [1, 44]. In this setup, we let the control space of each player $i$ be multi-dimensional, namely, $\Theta_i \equiv \mathbb{R}^5$, $i \in \mathcal{N}$, and, as in the previous example, the representation map of each player is instantiated by some preconfigured differentiable MLP whose output $x_i := \chi_i(\theta_i)$ is guaranteed to lie in $[0, 1]$. The MLP's output is going to be the probability with which player $i$ visits the El Farol bar. For the interested reader, further details, including technical specification of the MLPs, and loss functions of the games, can be found in the supplementary material. Figure 5 provides a comparative analysis of

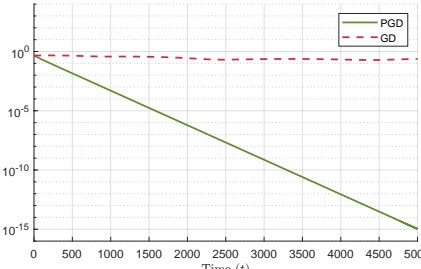 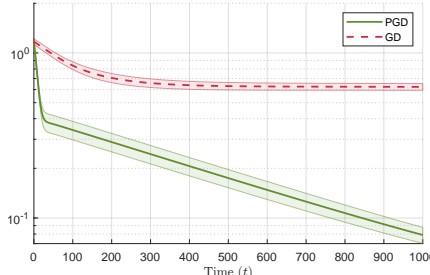

**Figure 5:** The evolution of the $L^2$ error function $\mathrm{Err}_\mathcal{X}(\chi(\theta))$ of (PHGD) and gradient descent (GD) with constant step-size $0.01$ in the regularized games of Matching Pennies and El Farol Bar as depicted in a semi-logarithmic scale. In the Matching Pennies game (left) we depict a single trajectory initialized at $(1.25, 2.25)$, while in the El Farol Bar game (right) game we depict the mean and confidence bounds of $100$ random trajectories.

the performance between (PHGD), and the standard GD method in the aforementioned two game scenarios. In Figure 5 (left) we explore the Matching Pennies game, where GD exhibits slightly erratic behavior. Despite eventually converging to the game's equilibrium point, GD's convergence rate, in this case, can be described as linear at best. In Figure 5 (right), we examine the El Farol Bar game. Interestingly, in this highly complex setup, GD fails to reach the equilibrium point entirely. In contrast, (PHGD) not only manages to converge in both of these setups, but it also consistently maintains an exponential rate of convergence. This stark difference underscores the efficacy and robustness of our algorithm.

# 6 Conclusion

This paper proposed a new algorithmic framework with strong formal convergence guarantees in a general class non-convex games with a latent monotone structure. Our algorithmic method – which we call preconditioned hidden gradient descent – relies on an appropriately chosen gradient preconditioning scheme akin to natural gradient ideas. Our class of games combines the useful structure of monotone operators as well as the notion of latent/hidden variables that arise in neural networks and can thus model numerous AI applications. Our results indicate the possibility of deep novel algorithmic ideas emerging at the intersection of game theory, non-convex optimization and ML and offers exciting directions for future work.

## Acknowledgments and Disclosure of Funding

This research has been partially supported by the National Research Foundation, Singapore and DSO National Laboratories under its AI Singapore Program (AISG Award No: AISG2-RP-2020-016), grant PIESGP-AI-2020-01, AME Programmatic Fund (Grant No.A20H6b0151) from A*STAR and Provost's Chair Professorship grant RGEPPV2101, project MIS 5154714 of the National Recovery and Resilience Plan Greece 2.0 funded by the European Union under the NextGenerationEU Program, and the French National Research Agency (ANR) in the framework of the PEPR IA FOUNDRY project (ANR-23-PEIA-0003), the "Investissements d'avenir" program (ANR-15-IDEX-02), the LabEx PERSYVAL (ANR-11-LABX-0025-01), and MIAI@Grenoble Alpes (ANR-19-P3IA-0003). PM is also with the Archimedes Research Unit – Athena RC – University of Athens.

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

## Organization of the appendix

## A  Omitted proofs from Section 3

In this first appendix, we provide the technical proofs of our results for the continuous-time dynamics (PHGF), namely Lemma 1 and Proposition 1. For convenience, we restate the relevant results as needed.

**Lemma 1.** *Under the dynamics* (7), *we have*

$$\dot{E}(\theta) = -\sum\nolimits_{i \in \mathcal{N}} [g_i(\chi(\theta))]^\intercal \mathbf{J}_i(\theta_i) \mathbf{P}_i(\theta_i) [\mathbf{J}_i(\theta_i)]^\intercal (\chi_i(\theta_i) - x_i^*) \tag{9}$$

*where, as per Section 2, $\mathbf{J}_i(\theta_i) \in \mathbb{R}^{d_i \times m_i}$ denotes the Jacobian matrix of the map $\chi_i \colon \Theta_i \to \mathcal{X}_i$. More compactly, letting $\mathbf{P} \coloneqq \bigoplus_i \mathbf{P}_i$ denote the block-diagonal ensemble of the players' individual preconditioning matrices $\mathbf{P}_i$ (and suppressing control variable arguments for concision), we have*

$$\dot{E} = -g(x)^\intercal \cdot \mathbf{J} \mathbf{P} \mathbf{J}^\intercal \cdot (x - x^*) \tag{10}$$

*Proof.* Observe that if we expand the dynamical system of equations in (PHGF) with respect to each individual coordinate, we have that for each player $i \in \mathcal{N}$, for all profiles $\theta \in \Theta$, and for all coordinates $\alpha \in \{1, \dots, m_i\}$:

$$\dot{\theta}_{i\alpha} = \sum_{\beta=1}^{m_i} \mathbf{P}_{i\alpha\beta}(\theta_i) \frac{\partial \ell_i(\theta)}{\partial \theta_{i\beta}}. \tag{A.1}$$

Now, since $\Gamma$ has a hidden monotone structure, and due to the decomposition $\ell_i(\theta) = f_i(x)$, we can expand the above as

$$\dot{\theta}_{i\alpha} = \sum_{\beta=1}^{m_i} \mathbf{P}_{i\alpha\beta}(\theta_i) \sum_{r=1}^{d_i} \frac{\partial f_i(x)}{\partial x_{ir}} \frac{\partial x_{ir}}{\partial \theta_{i\beta}}. \tag{A.2}$$

Furthermore, if we expand the left-hand side (LHS) of (9) we also get

$$\dot{E}(\theta) = \sum_{i=1}^{N} \sum_{\alpha=1}^{m_i} \frac{\partial E(\theta)}{\partial \theta_{i\alpha}} \dot{\theta}_{i\alpha}, \tag{A.3}$$

where each of the summands of the above expression can also be expanded further to

$$\frac{\partial E(\theta)}{\partial \theta_{i\alpha}} \dot{\theta}_{i\alpha} = -\sum_{l=1}^{d_i} (x_{il} - x_{il}^*) \frac{\partial x_{il}}{\partial \theta_{i\alpha}} \dot{\theta}_{i\alpha}. \tag{A.4}$$

Putting everything together (9) follows by some trivial substitutions. ∎

**Proposition 1.** *Suppose that $\Gamma$ admits a latent strictly monotone structure in the sense of* (3b)*. Then the energy function* (5) *is a strict Lyapunov function for* (PHGF)*, and every limit point $\theta^*$ of $\theta(t)$ is a Nash equilibrium of $\Gamma$.*

*Proof.* First, observe that for any player $i \in \mathcal{N}$, and $\theta_i \in \Theta_i$, since the player's representation map $\chi_i$ is faithful, i.e., $\mathbf{J}_i(\theta_i)$ is maximal rank, it holds:

$$\begin{aligned}
\mathbf{J}_i(\theta_i)\mathbf{P}_i(\theta_i)\mathbf{J}_i(\theta_i)^\mathsf{T} &= \mathbf{J}_i(\theta_i)\mathbf{J}_i(\theta_i)^+ \left[\mathbf{J}_i(\theta_i)^+\right]^\mathsf{T} \mathbf{J}_i(\theta_i)^\mathsf{T} \\
&= \left[\mathbf{J}_i(\theta_i)\mathbf{J}_i(\theta_i)^+\right]^\mathsf{T} \\
&= \mathbf{I}.
\end{aligned} \tag{A.5}$$

That is, $\sum_{\alpha=1}^{m_i} \sum_{\beta=1}^{m_i} \frac{\partial x_{il}}{\partial \theta_{i\alpha}} \frac{\partial x_{ir}}{\partial \theta_{i\beta}} \mathbf{P}_{i\alpha\beta}(\theta_i) = \delta_{lr}$ for all $l, r \in \{1, \dots, d_i\}$, where $\delta_{lr}$ is the Kronecker delta.

Now, by Lemma 1 we have that:

$$\begin{aligned}
\dot{E}(\theta) &= -\sum_{i=1}^{N} \sum_{l=1}^{d_i} \sum_{r=1}^{d_i} \delta_{lr}(x_{il} - x_{il}^*) \frac{\partial f_i(x)}{\partial x_{ir}} \\
&= -\sum_{i=1}^{N} \sum_{l=1}^{d_i} (x_{il} - x_{il}^*) \frac{\partial f_i(x)}{\partial x_{il}} \\
&= -\langle g(x), x - x^* \rangle.
\end{aligned} \tag{A.6}$$

which is negative, since $x^*$ is an optimizer of $f$, i.e., it satisfies the (MVI) due to monotonicity of $g$. Additionally, since $g$ is strictly monotone, $x^*$ is the unique optimizer of $f$, and, therefore, the above condition holds with equality if and only if $x = x^*$. ∎

# B   Auxiliary results from Section 4

**B.1. Convergence metrics and merit functions.**   In this appendix, we provide the technical scaffolding required for the analysis of (PHGD), namely Lemmas 2–4 in Section 3.2. As before, we restate the relevant results as needed.

**Lemma 2.** *Suppose that $g$ is monotone. If $\hat{x}$ is a solution of* (SVI)/(MVI)*, we have $\mathrm{Gap}_{\mathcal{C}}(\hat{x}) = 0$ whenever $\hat{x} \in \mathcal{C}$. Conversely, if $\mathrm{Gap}_{\mathcal{C}}(\hat{x}) = 0$ and $\mathcal{C}$ is a neighborhood of $\hat{x}$ in $\mathcal{X}$, then $\hat{x}$ is a solution of* (SVI)/(MVI)*.*

*Proof of Lemma 2.* Let $\hat{x} \in \mathcal{X}$ be a solution of (SVI)/(MVI) so $\langle g(\hat{x}), x - \hat{x} \rangle \geq 0$ for all $x \in \mathcal{X}$. Then, by the monotonicity of $g$, we get:

$$\begin{aligned}
\langle g(x), \hat{x} - x \rangle &\leq \langle g(x) - g(\hat{x}), \hat{x} - x \rangle + \langle g(\hat{x}), \hat{x} - x \rangle \\
&= -\langle g(\hat{x}) - g(x), \hat{x} - x \rangle - \langle g(\hat{x}), x - \hat{x} \rangle \leq 0,
\end{aligned} \tag{B.1}$$

so $\mathrm{Gap}_{\mathcal{C}}(\hat{x}) \leq 0$. On the other hand, if $\hat{x} \in \mathcal{C}$, we also get $\mathrm{Gap}(\hat{x}) \geq \langle g(\hat{x}), \hat{x} - \hat{x} \rangle = 0$, so we conclude that $\mathrm{Gap}_{\mathcal{C}}(\hat{x}) = 0$.

For the converse statement, assume that $\mathrm{Gap}_{\mathcal{C}}(\hat{x}) = 0$ for some $\hat{x} \in \mathcal{C}$ and suppose that $\mathcal{C}$ contains a neighborhood of $\hat{x}$ in $\mathcal{X}$. We then claim that $\hat{x}$ is a solution of (MVI) over $\mathcal{C}$, i.e.,

$$\langle g(x), x - \hat{x} \rangle \geq 0 \quad \text{for all } x \in \mathcal{C}. \tag{B.2}$$

To see this, assume to the contrary that there exists some $x_1 \in \mathcal{C}$ such that

$$\langle g(x_1), x_1 - \hat{x} \rangle < 0 \tag{B.3}$$

so, in turn, we get

$$0 = \mathrm{Gap}_{\mathcal{C}}(\hat{x}) \geq \langle g(x_1),\, \hat{x} - x_1 \rangle > 0, \tag{B.4}$$

a contradiction.

With this intermediate "local" result in hand, we are now in a position to prove that $\hat{x}$ solves (SVI). Indeed, if we suppose to the contrary that there exists some $z_1 \in \mathcal{X}$ such that $\langle g(\hat{x}),\, z_1 - \hat{x} \rangle < 0$, then, by the continuity of $g$, there exists a neighborhood $\mathcal{U}'$ of $\hat{x}$ in $\mathcal{X}$ such that

$$\langle g(x),\, z_1 - x \rangle < 0 \quad \text{for all } x \in \mathcal{U}'. \tag{B.5}$$

Hence, assuming without loss of generality that $\mathcal{U}' \subset \mathcal{U} \subset \mathcal{C}$ (the latter assumption due to the assumption that $\mathcal{C}$ contains a neighborhood of $\hat{x}$), and taking $\lambda > 0$ sufficiently small so that $x = \hat{x} + \lambda(z_1 - \hat{x}) \in \mathcal{U}'$, we get that $\langle g(x),\, x - \hat{x} \rangle = \lambda \langle g(x),\, z_1 - \hat{x} \rangle < 0$, in contradiction to (B.2). We thus conclude that $\hat{x}$ is a solution of (SVI) – and hence, by monotonicity, also of (MVI). $\blacksquare$

For intuition, we discuss below some other merit functions that could be considered as valid convergence metrics. In this regard, the first thing to note is that the definition of $\mathrm{Gap}(\hat{\theta})$ effectively goes through the game's *latent space*, so it is natural to ask whether a similar merit function can be defined directly on the game's *control space*. To do so, it will be more convenient to start with a linearized variant of $\mathrm{Gap}_{\mathcal{X}}$ defined over the cone of tangent directions at $\hat{x}$, namely the so-called "tangent residual gap"

$$\mathrm{TGap}_{\mathcal{X}}(\hat{x}) \coloneqq - \min_{z \in \mathrm{TC}_{\mathcal{X}}(\hat{x}), \|z\| \leq 1} \langle g(\hat{x}),\, z \rangle \tag{B.6}$$

i.e., the maximum "ascent" step $\langle -g(x),\, z \rangle$ over all admissible displacement directions $z$ from $\hat{x}$.[4] Just like $\mathrm{Gap}_{\mathcal{C}}$, this linearized merit function correctly identifies solutions of (SVI)/(MVI):

**Lemma B.1.** *For all $\hat{x} \in \mathcal{X}$, we have $\mathrm{TGap}_{\mathcal{X}}(\hat{x}) \geq 0$ with equality if and only if $\hat{x}$ solves* (SVI)/(MVI).

*Proof.* Let $\hat{x} \in \mathcal{X}$ be some arbitrary profile of latent variables. By definition, we have that $\mathrm{TGap}_{\mathcal{X}}(\hat{x}) = - \min_{z \in \mathrm{TC}_{\mathcal{X}}(\hat{x}), \|z\| \leq 1} \langle g(\hat{x}),\, z \rangle$, where $\mathrm{TC}_{\mathcal{X}}(\hat{x})$ is the tangent cone to $\mathcal{X}$ at $\hat{x}$. Observe that since $\mathrm{TC}_{\mathcal{X}}(\hat{x})$ is a cone, $0 \in \mathrm{TC}_{\mathcal{X}}(\hat{x})$, so $\mathrm{TGap}_{\mathcal{X}}(\hat{x}) \geq -\langle g(\hat{x}),\, 0 \rangle = 0$. Now assume that $\mathrm{TGap}_{\mathcal{X}}(\hat{x}) = 0$. Then, the following are equivalent:

$$\begin{aligned}
\mathrm{TGap}_{\mathcal{X}}(\hat{x}) = 0 &\iff - \min_{z \in \mathrm{TC}_{\mathcal{X}}(\hat{x}), \|z\| \leq 1} \langle g(\hat{x}),\, z \rangle = 0 \\
&\iff \langle g(\hat{x}),\, z \rangle \geq 0 \quad \text{for all } z \in \mathrm{TC}_{\mathcal{X}}(\hat{x}), \|z\| \leq 1 \\
&\iff \langle g(\hat{x}),\, z \rangle \geq 0 \quad \text{for all } z \in \mathrm{TC}_{\mathcal{X}}(\hat{x}),
\end{aligned} \tag{B.7}$$

where the last equivalence follows because $\mathrm{TC}_{\mathcal{X}}(\hat{x})$ is a cone. Rearranging the terms, we may equivalently write $\langle -g(\hat{x}),\, z \rangle \leq 0$ for all $z \in \mathrm{TC}_{\mathcal{X}}(\hat{x})$. Notice that, by the definition of the tangent cone, the latter is equivalent to $\langle -g(\hat{x}),\, \hat{x} - z \rangle \leq 0$ for all $z \in \mathcal{X}$. Therefore, $\mathrm{TGap}_{\mathcal{X}}(\hat{x}) = 0$ if and only if $\hat{x}$ satisfies (SVI). $\blacksquare$

Now, if $\theta^*$ is a Nash equilibrium of the base game, the latent variable configuration $x^* = \chi(\theta^*)$ solves (SVI)/(MVI), so $\mathrm{TGap}_{\mathcal{X}}$ is a valid equilibrium convergence metric for $\Gamma$. However, since $\mathrm{TGap}_{\mathcal{X}}$ is still defined on the game's latent space, it does not give a straightforward way of defining the quality of a candidate solution directly on the game's control space. To that end, one could consider the gap function

$$\mathrm{TGap}_{\Theta}(\hat{\theta}) \coloneqq - \min_{\eta \in \mathbb{R}^m, \|\eta\| \leq 1} \langle v(\hat{\theta}),\, \eta \rangle = \|v(\hat{\theta})\|_* \tag{B.8}$$

where $v(\theta) \coloneqq (\nabla_i \ell_i(\theta))_{i \in \mathcal{N}}$ collects the players' loss gradients relative to their control variables, and $\|\cdot\|_*$ denotes its dual norm (so $\mathrm{TGap}_{\Theta}(\hat{\theta})$ is evaluated *directly* on the game's control space). Of course, as control variables are mapped to latent variables, $\chi$ introduces a certaint distortion (due to nonlinearities). This distortion can be quantified by the following lemma (see also Fig. 6 above):

**Lemma B.2.** *Let $\hat{x} = \chi(\hat{\theta})$ for some $\hat{\theta} \in \Theta$. We then have*

$$\sigma_{\min} \mathrm{TGap}_{\mathcal{X}}(\hat{x}) \leq \mathrm{TGap}_{\Theta}(\hat{\theta}) \leq \sigma_{\max} \mathrm{TGap}_{\mathcal{X}}(\hat{x}) \tag{B.9}$$

*In particular, $\mathrm{TGap}_{\Theta}(\hat{\theta}) \geq 0$ for all $\hat{\theta} \in \Theta$, with equality if and only if $\hat{\theta}$ is a Nash equilibrium of $\Gamma$.*

---

[4]In the above, $\mathrm{TC}_{\mathcal{X}}(\hat{x})$ denotes the tangent cone to $\mathcal{X}$ at $\hat{x}$, that is, the closure of the set of all rays emanating from $\hat{x}$ and intersecting $\mathcal{X}$ in at least one other point.

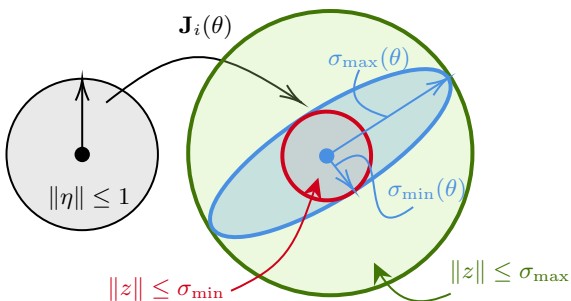

**Figure 6:** The distortion of a ball by $\mathbf{J}(\theta)$.

*Proof.* To begin with, let us define the balls $\mathbb{B}_\Theta \equiv \{\eta \in \mathbb{R}^m : \|\eta\| \leq 1\}$, and $\mathbb{B}_{\mathcal{X}} \equiv \{z \in \mathbb{R}^d : \|z\| \leq 1\}$. Now, let $\hat{\theta} \in \Theta$ be some arbitrary profile of control variables. By Assumption 2, we have that the singular values of the Jacobian $\mathbf{J}(\hat{\theta})$ of the representation map $\chi$ are bounded from above and below by $\sigma_{\max} < \infty$ and $\sigma_{\min} > 0$ respectively. That is, $\sigma_{\min}\|\eta\| \leq \|\mathbf{J}(\hat{\theta})\eta\| \leq \sigma_{\max}\|\eta\|$ for all $\eta \in \mathbb{R}^m$, which implies that $\sigma_{\min}\mathbb{B}_{\mathcal{X}} \subseteq \mathbf{J}(\hat{\theta})\mathbb{B}_\Theta \subseteq \sigma_{\max}\mathbb{B}_{\mathcal{X}}$. By definition, we also have that, $v(\hat{\theta}) = \mathbf{J}(\hat{\theta})^\intercal g(\chi(\hat{\theta})) = \mathbf{J}(\hat{\theta})^\intercal g(\hat{x})$, and, hence

$$\mathrm{TGap}_\Theta(\hat{\theta}) = -\min_{\eta \in \mathbb{B}_\Theta} \langle v(\hat{\theta}),\, \eta \rangle = -\min_{z \in \mathbf{J}(\hat{\theta})\mathbb{B}_\Theta} \langle g(\hat{x}),\, z \rangle \tag{B.10}$$

Now, recall that the set of Nash equilibria $\Theta^* \subset \Theta$ is non-empty. In particular, since the $\Theta \equiv \mathbb{R}^m$ is an open set, the solution set lies in an open set, which, in conjunction with the fact that the representation maps are faithful representations of $\Theta$ to $\mathcal{X}$, also implies that $\chi(\Theta^*)$ lies in the interior of $\mathcal{X}$. Finally, since $\chi$ is smooth, and in conjunction with the previous observations, it follows that $\mathrm{TGap}_\Theta(\hat{\theta}) = \|v(\hat{\theta})\|$ and $\mathrm{TGap}_{\mathcal{X}}(\hat{x}) = \|g(\hat{x})\|$, hence it is easy to see from the above discussion that

$$\sigma_{\min}\, \mathrm{TGap}_{\mathcal{X}}(\chi(\hat{\theta})) \leq \mathrm{TGap}_\Theta(\hat{\theta}) \leq \sigma_{\max}\, \mathrm{TGap}_{\mathcal{X}}(\chi(\hat{\theta})) \tag{B.11}$$

and our proof is complete. ∎

In view of the above discussion, the reader may wonder why not use the tangent gap $\mathrm{TGap}_\Theta$ directly as a performance metric. The reason for this is that, if the solutions $x^*$ of (SVI)/(MVI) do not belong to the image of $\chi$, the dual norm of $v(\hat{\theta})$ may be too stringent as a performance metric as it does not vanish near an equilibrium (e.g., think of the operator $g(x) = x$ for $x$ between 0 and 1). In this case, equilibria can be approximated to arbitrary accuracy but never attained, so the latent gap functions $\mathrm{Gap}$ and $\mathrm{Err}$ are more appropriate as performance measures.

### B.2. Convergence analysis and template inequalities.

**Lemma 3.** *Fix some $\hat{x} \in \mathcal{X}$, and consider the energy function $E(\theta; \hat{x}) = (1/2)\|\chi(\theta) - \hat{x}\|^2$. Then, for all $\theta \in \Theta$, we have*

$$\mathbf{J}(\theta)\mathbf{P}(\theta)\nabla_\theta E(\theta; \hat{x}) = \chi(\theta) - \hat{x}. \tag{20}$$

*Proof.* Let $i \in \mathcal{N}$ be some arbitrary player, let $\theta \in \Theta$ be some arbitrary profile of control variables, and let $\eta \in \mathbb{R}^{d_i}$ be some arbitrary vector. Then for each coordinate $\alpha \in \{1, \dots, m_i\}$, and latent profile $\hat{x} \in \mathcal{X}$, we have that

$$\frac{\partial E(\theta; \hat{x})}{\partial \theta_{i\alpha}} = \sum_{j=1}^{d_i} (x_{ij} - \hat{x}_{ij}) \frac{\partial x_{ij}}{\partial \theta_{i\alpha}}. \tag{B.12}$$

That is $\nabla_{\theta_i} E(y; \hat{x}) = \mathbf{J}_i(\theta_i)^\intercal (x_i - \hat{x}_i)$. Now, we can multiply both sides of the above expression with $\mathbf{P}_i(\theta_i)$ to get

$$\begin{aligned}
\mathbf{P}_i(\theta_i)\, \nabla_{\theta_i} E(\theta; \hat{x}) &= \mathbf{P}_i(\theta_i)\mathbf{J}_i(\theta_i)^\intercal (x_i - \hat{x}_i) \\
&= \mathbf{J}_i(\theta_i)^+ \big(\mathbf{J}_i(\theta_i)^+\big)^\intercal \mathbf{J}_i(\theta_i)^\intercal (x_i - \hat{x}_i) \\
&= \mathbf{J}_i(\theta_i)^+ (x_i - \hat{x}_i),
\end{aligned} \tag{B.13}$$

where in the last equality we used the fact that the $i$-th player's representation map $\chi_i$ is a faithful representation, i.e., $\mathbf{J}_i(\theta_i)$ is maximal rank. Finally, by expanding the LHS of (20), and applying the above simplification the result follows. ∎

**Lemma 4** (Template inequality). *Suppose that Assumptions 1 and 2 hold. Then, with notation as in Lemma 3, the sequence $E_t := (1/2)\|\chi(\theta_t) - \hat{x}\|^2$, $t = 1, 2, \ldots$, satisfies*

$$E_{t+1} \leq E_t - \gamma_t g(x_t)^\mathsf{T}(x_t - \hat{x}) + \gamma_t \phi_t + \gamma_t^2 \psi_t, \tag{21}$$

*where $x_t := \chi(\theta_t)$, $\phi_t := (\mathbf{J}(\theta_t)\mathbf{P}(\theta_t)V_t - g(x_t))^\mathsf{T}(x_t - \hat{x})$ and $\psi_t$ is a random error sequence with $\sup_t \mathbb{E}[\psi_t] < \infty$.*

*Proof.* For each $t = 1, 2, \ldots$ we expand $x_{t+1}$ using its second-order Taylor approximation at $\theta_t$; i.e., for each player $i = 1, \ldots, N$ and each coordinate $l = 1, \ldots, d_i$:

$$
\begin{aligned}
x_{il,t+1} &= \chi_{il}(\theta_{t+1}) \\
&= \chi_{il}(\theta_{i,t}) + \gamma_t \langle \nabla \chi_{il}(\theta_{i,t}), \theta_{i,t+1} - \theta_{i,t} \rangle \\
&\quad + \gamma_t^2 (\theta_{i,t+1} - \theta_{i,t})^\mathsf{T} H_{il}(\tilde{\theta}_{i,t})(\theta_{i,t+1} - \theta_{i,t}),
\end{aligned}
\tag{B.14}
$$

where $H_{il}(\tilde{\theta}_{i,t})$ is the Hessian of the latent map $\chi_{il}$ at some convex combination $\tilde{\theta}_{i,t}$ of $\theta_{i,t}$ and $\theta_{i,t+1}$. Then, further expanding $\theta_{t+1}$, we also get

$$x_{il,t+1} = x_{il,t} - \gamma_t \langle \nabla \chi_{il}(\theta_{i,t}), \mathbf{P}_{i,t}V_{i,t} \rangle + \gamma_t^2 (\mathbf{P}_{i,t}V_{i,t})^\mathsf{T} H_{il}(\tilde{\theta}_{i,t})(\mathbf{P}_{i,t}V_{i,t}). \tag{B.15}$$

Plugging the above expansion to $E(\theta_{t+1}; \hat{x})$ for arbitrary state $\hat{x}$ we get

$$
\begin{aligned}
E_{t+1} &= E(\theta_{t+1}; \hat{x}) \\
&= \frac{1}{2} \sum_{i=1}^N \sum_{l=1}^{d_i} (x_{t+1} - \hat{x})^2 \\
&= \frac{1}{2} \sum_{i=1}^N \sum_{l=1}^{d_i} [x_{il,t} - \hat{x}_{il} - \gamma_t \langle \nabla \chi_{il}(\theta_{i,t}), \mathbf{P}_{i,t}V_{i,t} \rangle \\
&\quad + \gamma_t^2 (\mathbf{P}_{i,t}V_{i,t})^\mathsf{T} H_{il}(\tilde{\theta}_{i,t})(\mathbf{P}_{i,t}V_{i,t})]^2 \\
&= E_t - \gamma_t \sum_{i=1}^N \sum_{l=1}^{d_i} \langle \nabla \chi_{il}(\theta_{i,t}), \mathbf{P}_{i,t}V_{i,t} \rangle (x_{il,t} - \hat{x}_{il}) + \gamma_t^2 \eta_t \\
&= E_t - \gamma_t (\mathbf{J}_t \mathbf{P}_t V_t)^\mathsf{T}(x_t - \hat{x}) + \gamma_t^2 \eta_t,
\end{aligned}
\tag{B.16}
$$

where the second-order term $\eta_t$ is given by the formula

$$
\begin{aligned}
\eta_t &= \sum_{i=1}^N \sum_{l=1}^{d_i} \langle \nabla \chi_{il}(\theta_{i,t}), \mathbf{P}_{i,t}V_{i,t} \rangle^2 \\
&\quad + \sum_{i=1}^N \sum_{l=1}^{d_i} (\mathbf{P}_{i,t}V_{i,t})^\mathsf{T} H_{il}(\tilde{\theta}_{i,t})(\mathbf{P}_{i,t}V_{i,t})(x_{il,t} - \hat{x}_{il}) \\
&\quad - \gamma_t \sum_{i=1}^N \sum_{l=1}^{d_i} \langle \nabla \chi_{il}(\theta_{i,t}), \mathbf{P}_{i,t}V_{i,t} \rangle (\mathbf{P}_{i,t}V_{i,t})^\mathsf{T} H_{il}(\tilde{\theta}_{i,t})(\mathbf{P}_{i,t}V_{i,t}) \\
&\quad + \gamma_t^2 \sum_{i=1}^N \sum_{l=1}^{d_i} [(\mathbf{P}_{i,t}V_{i,t})^\mathsf{T} H_{il}(\tilde{\theta}_{i,t})(\mathbf{P}_{i,t}V_{i,t})]^2.
\end{aligned}
\tag{B.17}
$$

For the second-order term, observe that by Assumption 2, we have that $[\mathbf{J}_t^\mathsf{T}\mathbf{J}_t]^+ \preceq \frac{1}{\sigma_{\min}^2}\mathbf{I}$. Using that fact, it follows that

$$
\begin{aligned}
\sum_{i=1}^{N}\sum_{l=1}^{d_i}\langle\nabla\chi_{il}(\theta_{i,t}),\mathbf{P}_{i,t}V_{i,t}\rangle^2 &= (\mathbf{J}_t\mathbf{P}_tV_t)^\mathsf{T}\mathbf{J}_t\mathbf{P}_tV_t \\
&= V_t^\mathsf{T}[\mathbf{J}_t^\mathsf{T}\mathbf{J}_t]^+\mathbf{J}_t^\mathsf{T}\mathbf{J}_t[\mathbf{J}_t^\mathsf{T}\mathbf{J}_t]^+V_t \\
&= V_t^\mathsf{T}[\mathbf{J}_t^\mathsf{T}\mathbf{J}_t]^+V_t \\
&\leq \frac{1}{\sigma_{\min}^2}\|V_t\|^2,
\end{aligned}
\tag{B.18}
$$

Furthermore, since the representation maps $\chi_i$ are $\beta$-Lipschitz smooth for some Lipschitz modulus $\beta$, we have that $H_{i,l}(\tilde{\theta}_{i,t}) \preceq \beta\mathbf{I}$ for each player $i$ and coordinate $l$. Consequently, we have that

$$
(\mathbf{P}_{i,t}V_{i,t})^\mathsf{T}H_{il}(\tilde{\theta}_{i,t})(\mathbf{P}_{i,t}V_{i,t}) \leq \beta V_{i,t}^\mathsf{T}\mathbf{P}_{i,t}^2V_{i,t} \leq \frac{\beta}{\sigma_{\min}^4}\|V_{i,t}\|^2 \leq \frac{\beta}{\sigma_{\min}^4}\|V_t\|^2.
\tag{B.19}
$$

Let $D = \operatorname{diam}(\chi)$, then, by applying the Cauchy-Schwarz inequality, we also get that

$$
\eta_t \leq \frac{1}{\sigma_{\min}^2}\|V_t\|^2 + \frac{dD\sqrt{\beta}}{\sigma_{\min}^2}\|V_t\| + \gamma\frac{\sqrt{\beta}}{\sigma_{\min}^3}\|V_t\|^2 + \gamma^2\frac{d\beta^2}{\sigma_{\min}^8}\|V_t\|^2,
\tag{B.20}
$$

where $\gamma = \sup_{t=1,2,\ldots}\gamma_t$, and $d = \sum_{i=1}^{N}d_i$.

Finally, let us consider the first-order term of (B.16). Let $v_t = (\nabla_i\ell_i(\theta_t))_{i\in\mathcal{N}}$, and $g_t = g(x_t)$. Then, by definition, we also have that $v_t = \mathbf{J}_t^\mathsf{T}g_t$. Moreover, recall that, by construction, $\mathbf{J}_t\mathbf{P}_t\mathbf{J}_t^\mathsf{T} = \mathbf{I}$. Consequently, we can write

$$
\begin{aligned}
(\mathbf{J}_t\mathbf{P}_tV_t)^\mathsf{T}(x_t-\hat{x}) &= (\mathbf{J}_t\mathbf{P}_tv_t)^\mathsf{T}(x_t-\hat{x}) + [\mathbf{J}_t\mathbf{P}_t(V_t-v_t)]^\mathsf{T}(x_t-\hat{x}) \\
&= (\mathbf{J}_t\mathbf{P}_t\mathbf{J}_t^\mathsf{T}g_t)^\mathsf{T}(x_t-\hat{x}) + [\mathbf{J}_t\mathbf{P}_t(V_t-\mathbf{J}_t^\mathsf{T}g_t)]^\mathsf{T}(x_t-\hat{x}) \\
&= g_t^\mathsf{T}(x_t-\hat{x}) + (\mathbf{J}_t\mathbf{P}_tV_t-g_t)^\mathsf{T}(x_t-\hat{x}),
\end{aligned}
\tag{B.21}
$$

which concludes the proof. ∎

## C  Proofs of Theorems 1–3

We are now in a position to prove Theorems 1–3 regarding the convergence properties of (PHGD). We proceed sequentially, restating the relevant results as needed.

**Theorem 1** (PHGD in hidden monotone games)**.** *Suppose that players run* (PHGD) *in a hidden monotone game with learning rate $\gamma_t \propto 1/t^{1/2}$. Then, under Assumptions 1 and 2, the averaged process $\bar{\theta}_t \in \chi^{-1}\big(t^{-1}\sum_{s=1}^{t}x_s\big)$ enjoys the equilibrium convergence rate*

$$
\mathbb{E}[\operatorname{Gap}(\bar{\theta}_t)] = \mathcal{O}(\log t/\sqrt{t}).
\tag{16}
$$

*Proof.* Let $\theta_1 \in \Theta$ be some arbitrary initialization of the algorithm, and let $\tilde{x} \in \mathcal{X}$ be some arbitrary profile of latent variables. Next, by Lemma 4, we have that for all $t \geq 1$:

$$
\begin{aligned}
E_{t+1} &\leq E_t - \gamma_t\langle g(x_t), x_t-\hat{x}\rangle + \gamma_t\phi_t + \gamma_t^2\psi_t \\
\phi_t &= (\mathbf{J}(\theta_t)\mathbf{P}(\theta_t)V_t - g(x_t))^\mathsf{T}(x_t-\hat{x}) \\
\psi_t &= \kappa\|V_t\|^2, \quad \text{for some } \kappa > 0.
\end{aligned}
\tag{C.1}
$$

Rearranging the terms, the above is equivalent to

$$
\gamma_t\langle g(x_t), x_t-\tilde{x}\rangle \leq E_t - E_{t+1} + \gamma_t\phi_t + \gamma_t^2\psi_t.
\tag{C.2}
$$

Furthermore, by the monotonicity of $g$, we also have that $\langle g(x_t) - g(\tilde{x}), x_t - \tilde{x}\rangle \geq 0$, and by combining the two, we get that for all $t \geq 1$:

$$
\gamma_t\langle g(\tilde{x}), x_t-\tilde{x}\rangle \leq \gamma_t\langle g(x_t), x_t-\tilde{x}\rangle \leq E_t - E_{t+1} + \gamma_t\phi_t + \gamma_t^2\psi_t.
\tag{C.3}
$$

Summing up the above terms in both sides of the inequality, we also get that

$$\sum_{s=1}^{t} \gamma_s \langle g(\tilde{x}), x_s - \tilde{x} \rangle \leq \sum_{s=1}^{t} \left[ E_s - E_{s+1} + \gamma_s \phi_s + \gamma_s^2 \psi_s \right]$$

$$= E_1 - E_{t+1} + \sum_{s=1}^{t} \gamma_s \phi_s + \sum_{s=1}^{t} \gamma_s^2 \psi_s \qquad \text{(C.4)}$$

$$\leq E_1 + \sum_{s=1}^{t} \gamma_s \phi_s + \sum_{s=1}^{t} \gamma_s^2 \psi_s.$$

Dividing all terms by $\tilde{\gamma}_t := \sum_{s=1}^{t} \gamma_s$, also yields

$$\langle g(\tilde{x}), \bar{x}_t - \tilde{x} \rangle = \left\langle g(\tilde{x}), \sum_{s=1}^{t} \frac{\gamma_s}{\tilde{\gamma}_t} x_s - \tilde{x} \right\rangle$$

$$= \sum_{s=1}^{t} \frac{\gamma_s}{\tilde{\gamma}_t} \langle g(\tilde{x}), x_s - \tilde{x} \rangle \qquad \text{(C.5)}$$

$$\leq \frac{E_1}{\tilde{\gamma}_t} + \frac{\sum_{s=1}^{t} \gamma_s \phi_s}{\tilde{\gamma}_t} + \frac{\sum_{s=1}^{t} \gamma_s^2 \psi_s}{\tilde{\gamma}_t},$$

where $\bar{x}_t := \sum_{s=1}^{t} \frac{\gamma_s}{\tilde{\gamma}_t} x_s$ is the time-average.

Next, let us define the following auxiliary process that will assist us in further bounding the above expression:

$$y_1 = x_1$$
$$y_{t+1} = \arg\min_{x \in \mathcal{X}} \|y_t - \gamma_t \eta_t - x\| \quad \text{for all } t \geq 2, \qquad \text{(C.6)}$$

where $\eta_t := \mathbf{J}(\theta_t) \mathbf{P}(\theta_t) V_t - g(x_t)$ for all $i \in \mathcal{N}$. Observe that

$$\gamma_t \phi_t = \gamma_t \langle \eta_t, x_t - \tilde{x} \rangle = \gamma_t \langle \eta_t, y_t - \tilde{x} \rangle + \gamma_t \langle \eta_t, x_t - y_t \rangle \quad \text{for all } t \geq 1. \qquad \text{(C.7)}$$

Let us, first, focus on the former of the two terms. Notice that for all $t \geq 1$, we can write

$$\gamma_t \langle \eta_t, y_t - \tilde{x} \rangle = \gamma_t \langle \eta_t, y_{t+1} - \tilde{x} \rangle + \gamma_t \langle \eta_t, y_t - y_{t+1} \rangle. \qquad \text{(C.8)}$$

Furthermore, by the optimality of $y_{t+1}$, $t \geq 2$, we have also have that

$$\langle y_{t+1} - y_t + \gamma_t \eta_t, y_{t+1} - \tilde{x} \rangle \leq 0. \qquad \text{(C.9)}$$

That is, $\gamma_t \langle \eta_t, y_{t+1} - \tilde{x} \rangle \leq \langle y_t - y_{t+1}, y_{t+1} - \tilde{x} \rangle$.

Let us also recall a couple of useful quadratic identities, namely, we have that $\|y_t - \tilde{x}\|^2 = \|y_t - y_{t+1} + y_{t+1} - \tilde{x}\|^2 = \|y_t - y_{t+1}\|^2 + 2\langle y_t - y_{t+1}, y_{t+1} - \tilde{x} \rangle + \|y_{t+1} - \tilde{x}\|^2$, and $\|\gamma_t \eta_t - (y_t - y_{t+1})\|^2 = \gamma_t^2 \|\eta_t\|^2 - 2\gamma_t \langle \eta_t, y_t - y_{t+1} \rangle + \|y_t - y_{t+1}\|^2$. Then, in conjunction with the above, we get that for all $t \geq 1$:

$$\gamma_t \langle \eta_t, y_t - \tilde{x} \rangle = \gamma_t \langle \eta_t, y_{t+1} - \tilde{x} \rangle + \gamma_t \langle \eta_t, y_t - y_{t+1} \rangle$$
$$\leq \langle y_t - y_{t+1}, y_{t+1} - \tilde{x} \rangle + \gamma_t \langle \eta_t, y_t - y_{t+1} \rangle$$
$$= \frac{1}{2} \|y_t - \tilde{x}\|^2 - \frac{1}{2} \|y_{t+1} - \tilde{x}\|^2 - \frac{1}{2} \|\gamma_t \eta_t - (y_t - y_{t+1})\|^2 + \frac{\gamma_t^2}{2} \|\eta_t\|^2 \qquad \text{(C.10)}$$
$$\leq \frac{1}{2} \|y_t - \tilde{x}\|^2 - \frac{1}{2} \|y_{t+1} - \tilde{x}\|^2 + \frac{\gamma_t^2}{2} \|\eta_t\|^2$$

Finally, summing both sides of the above inequalities, we also get that

$$
\sum_{s=1}^{t} \gamma_s \langle \eta_s,\, y_s - \tilde{x} \rangle \leq \frac{1}{2} \sum_{s=1}^{t} \|y_s - \tilde{x}\|^2 - \frac{1}{2} \sum_{s=1}^{t} \|y_{s+1} - \tilde{x}\|^2 + \frac{1}{2} \sum_{s=1}^{t} \gamma_s^2 \|\eta_s\|^2
$$

$$
= \frac{1}{2} \|y_1 - \tilde{x}\|^2 - \frac{1}{2} \|y_{t+1} - \tilde{x}\|^2 + \frac{1}{2} \sum_{s=1}^{t} \gamma_s^2 \|\eta_s\|^2
$$

$$
\leq \frac{1}{2} \|y_1 - \tilde{x}\|^2 + \frac{1}{2} \sum_{s=1}^{t} \gamma_s^2 \|\eta_s\|^2 \tag{C.11}
$$

$$
= \frac{1}{2} \|x_1 - \tilde{x}\|^2 + \frac{1}{2} \sum_{s=1}^{t} \gamma_s^2 \|\eta_s\|^2
$$

$$
= E_1 + \frac{1}{2} \sum_{s=1}^{t} \gamma_s^2 \|\eta_s\|^2
$$

With the above established, let us reconsider the quantity $\langle g(\tilde{x}),\, \bar{x}_t - \tilde{x} \rangle$. Specifically, due to the above derivations, we have that for all $t \geq 1$:

$$
\langle g(\tilde{x}),\, \bar{x}_t - \tilde{x} \rangle \leq \frac{E_1}{\tilde{\gamma}_t} + \frac{\sum_{s=1}^{t} \gamma_s \phi_s}{\tilde{\gamma}_t} + \frac{\sum_{s=1}^{t} \gamma_s^2 \psi_s}{\tilde{\gamma}_t}
$$

$$
= \frac{E_1}{\tilde{\gamma}_t} + \frac{1}{\tilde{\gamma}_t} \sum_{s=1}^{t} \gamma_t \langle \eta_t,\, y_t - \tilde{x} \rangle + \frac{1}{\tilde{\gamma}_t} \sum_{s=1}^{t} \gamma_t \langle \eta_t,\, x_t - y_t \rangle + \frac{\sum_{s=1}^{t} \gamma_s^2 \psi_s}{\tilde{\gamma}_t}
$$

$$
\leq \frac{3E_1}{2\tilde{\gamma}_t} + \frac{1}{2\tilde{\gamma}_t} \sum_{s=1}^{t} \gamma_s^2 \|\eta_s\|^2 + \frac{1}{\tilde{\gamma}_t} \sum_{s=1}^{t} \gamma_t \langle \eta_t,\, x_t - y_t \rangle + \frac{\kappa}{\tilde{\gamma}_t} \sum_{s=1}^{t} \gamma_s^2 \|V_s\|^2.
$$

Considering the mean of supremum over $\tilde{x} \in \mathcal{X}$ of the LHS, we end up with

$$
\mathbb{E}\left[ \sup_{\tilde{x} \in \mathcal{X}} \langle g(\tilde{x}),\, \bar{x}_t - \tilde{x} \rangle \right] \leq \frac{3E_1}{2\tilde{\gamma}_t} + \frac{1}{2\tilde{\gamma}_t} \sum_{s=1}^{t} \gamma_s^2 \, \mathbb{E}[\|\eta_s\|^2] + \frac{\kappa}{\tilde{\gamma}_t} \sum_{s=1}^{t} \gamma_s^2 \, \mathbb{E}[\|V_s\|^2]
$$

$$
\leq \frac{3E_1}{2\tilde{\gamma}_t} + \frac{M^2}{\tilde{\gamma}_t} \sum_{s=1}^{t} \gamma_s^2 + \frac{\kappa M^2}{\tilde{\gamma}_t} \sum_{s=1}^{t} \gamma_s^2
$$

$$
= \frac{3E_1}{2\tilde{\gamma}_t} + (1 + \kappa) \frac{M^2}{\tilde{\gamma}_t} \sum_{s=1}^{t} \gamma_s^2 \tag{C.12}
$$

$$
= \mathcal{O}\Big( \sum_{s=1}^{t} \gamma_s^2 / \tilde{\gamma}_t \Big)
$$

$$
= \mathcal{O}(\log t / \sqrt{t}),
$$

where the last inequality is direct consequence of the bounded second moment of the stochastic gradient in Assumption 1 ∎

**Theorem 2** (PHGD in hidden strongly monotone games). *Suppose that players run* (PHGD) *in a hidden $\mu$-strongly monotone game with $\gamma_t = \gamma/t$ for some $\gamma > \mu$. Then, under Assumptions 1 and 2, the induced sequence of play $\theta_t \in \Theta$, $t = 1, 2, \ldots$, enjoys the equilibrium convergence rate*

$$
\mathbb{E}[\mathrm{Err}_\Theta(\theta_t)] = \mathcal{O}(1/t). \tag{18}
$$

*Proof.* Let $\theta_1 \in \Theta$ be some arbitrary initialization of the algorithm, and let $\gamma \geq \frac{1}{2\mu}$ be arbitrary. Since the map $f$ is $\mu$-strongly monotone, we have, by the definition of strong monotonicity, that

$$
\langle g(x_t),\, x_t - x^* \rangle \geq \mu \|x - x^*\|^2 \quad \text{for all } x \in \mathcal{X}. \tag{C.13}
$$

Next, by Lemma 4, we have that for all $t \geq 1$:

$$
\begin{aligned}
E_{t+1} &\leq E_t - \gamma_t \langle g(x_t),\, x_t - \hat{x} \rangle + \gamma_t \phi_t + \gamma_t^2 \psi_t \\
\phi_t &= (\mathbf{J}(\theta_t)\mathbf{P}(\theta_t)V_t - g(x_t))^{\mathsf{T}}(x_t - \hat{x}) \\
\psi_t &= \kappa \|V_t\|^2, \quad \text{for some } \kappa > 0.
\end{aligned}
\tag{C.14}
$$

That is

$$
\begin{aligned}
E_{t+1} &\leq E_t - \mu\gamma_t \|x_t - x^*\|_2^2 + \gamma_t \phi_t + \gamma_t^2 \psi_t \\
&= E_t - 2\mu\gamma_t E_t + \gamma_t \phi_t + \gamma_t^2 \psi_t \\
&= (1 - 2\mu\gamma_t)E_t + \gamma_t \phi_t + \gamma_t^2 \psi_t.
\end{aligned}
\tag{C.15}
$$

Let $\mathcal{H}_t = \{\theta_s : s = 1, \ldots, t\}$ be the history of play up to iteration $t$. In expectation, it follows from the above that, for all $t \geq 1$:

$$
\begin{aligned}
\mathbb{E}[E_{t+1}] &\leq (1 - 2\mu\gamma_t)\,\mathbb{E}[E_t] + \gamma_t\,\mathbb{E}[\phi_t] + \gamma_t^2\,\mathbb{E}[\psi_t] \\
&= (1 - 2\mu\gamma_t)\,\mathbb{E}[E_t] + \gamma_t\,\mathbb{E}\Big[\mathbb{E}[\phi_t \mid \mathcal{H}_t]\Big] + \gamma_t^2\,\mathbb{E}\Big[\mathbb{E}[\psi_t \mid \mathcal{H}_t]\Big].
\end{aligned}
\tag{C.16}
$$

Note that $x_t$ is $\mathcal{H}_t$-measurable; hence, we have that

$$
\begin{aligned}
\mathbb{E}[\phi_t \mid \mathcal{H}_t] &= \big(\mathbf{J}(\theta_t)\mathbf{P}(\theta_t)\,\mathbb{E}[V_t \mid \mathcal{H}_t] - g(x_t)\big)^{\mathsf{T}}(x_t - \hat{x}) \\
&= \big(\mathbf{J}(\theta_t)\mathbf{P}(\theta_t)v(\theta_t) - g(x_t)\big)^{\mathsf{T}}(x_t - \hat{x}) \\
&= \big(\mathbf{J}(\theta_t)\mathbf{P}(\theta_t)\mathbf{J}(\theta_t)^{\mathsf{T}}g(x_t) - g(x_t)\big)^{\mathsf{T}}(x_t - \hat{x}) \\
&= \big(g(x_t) - g(x_t)\big)^{\mathsf{T}}(x_t - \hat{x}) \\
&= 0.
\end{aligned}
\tag{C.17}
$$

Furthermore, since, by Assumption 1, we have that $\mathbb{E}[\|V_t\|^2 \mid \mathcal{H}_t] \leq M^2$, it also holds that

$$
\mathbb{E}[\psi_t \mid \mathcal{H}_t] = \kappa\,\mathbb{E}[\|V_t\|^2 \mid \mathcal{H}_t] \leq \kappa M^2
\tag{C.18}
$$

Next, by some simple substitutions, we have

$$
\begin{aligned}
\mathbb{E}[E_{t+1}] &\leq (1 - 2\mu\gamma_t)\,\mathbb{E}[E_t] + \kappa M^2 \gamma_t^2 \\
&= \Big(1 - \frac{2\mu\gamma}{t}\Big)\,\mathbb{E}[E_t] + \frac{\kappa M^2 \gamma^2}{t^2}
\end{aligned}
\tag{C.19}
$$

Finally, since $\gamma > \frac{1}{2\mu}$, we may apply Chung's lemma [8] to get that, for all $t \geq 1$:

$$
\mathbb{E}[E_t] \leq \frac{\kappa M^2 \gamma^2}{2\mu\gamma - 1} \cdot \frac{1}{t} + \mathcal{O}\Big(\frac{1}{t^2} + \frac{1}{t^{2\mu\gamma}}\Big).
\tag{C.20}
$$

∎

**Theorem 3** (PHGD with full gradient feedback in hidden strongly monotone games)**.** *Suppose that players run* (PHGD) *in a hidden $\mu$-strongly monotone game with full gradient feedback, and a sufficiently small learning rate $\gamma > 0$. Then, under Assumptions 1 and 2, the induced sequence of play $\theta_t \in \Theta$, $t = 1, 2, \ldots$, converges to equilibrium at a geometric rate, i.e.,*

$$
\mathrm{Err}_\Theta(\theta_t) = \mathcal{O}(\rho^t)
\tag{19}
$$

*for some constant $\rho \in (0, 1)$ that depends only on the primitives of $\Gamma$ and the representation map $\chi$.*

*Proof.* Due to the absence of randomness, and therefore the absence of noise, Lemma 4 may be simplified to

$$
E_{t+1} \leq E_t - \gamma_t \langle g(x_t),\, x_t - x^* \rangle + \kappa \gamma_t^2 \|v_t\|^2
\tag{C.21}
$$

for some constants $\kappa > 0$, where $v_t := (\nabla_i \ell_i(\theta_t))_{i \in \mathcal{N}}$.

Let $\beta$ be the modulus of Lipschitz continuity of the gradients $v_t = \mathbf{J}(\theta_t)^{\mathsf{T}}g(x_t)$, and let $\frac{\mu}{2}$ be the modulus of the strong monotonicity of $g$. Recall that, by Assumption 2, we have that the singular

values of the Jacobian $\mathbf{J}(\theta)$ of the representation map $\chi$ are bounded from above and below by $\sigma_{\max} < \infty$ and $\sigma_{\min} > 0$ respectively. Therefore, since $v_t$ is $\beta$-Lipschitz, it follows that

$$\|v_t - v^*\|^2 \le \beta^2 \|\theta_t - \theta^*\|^2 \le \frac{\beta^2}{\sigma_{\min}^2} \|x_t - x^*\|^2 \tag{C.22}$$

In conjunction to the above, we then also have

$$
\begin{aligned}
E_{t+1} &\le E_t - \gamma_t \langle g(x_t),\, x_t - x^* \rangle + \kappa \gamma_t^2 \|v_t\|^2 \\
&= E_t - \gamma_t \langle g(x_t),\, x_t - x^* \rangle + \kappa \gamma_t^2 \|v_t - v^*\|^2 \\
&\le E_t - \mu \gamma_t \frac{1}{2} \|x_t - x^*\|^2 + \frac{\kappa \gamma_t^2 \beta^2}{\sigma_{\min}^2} \|x_t - x^*\|^2 \\
&= E_t - \mu \gamma_t E_t + \frac{2\kappa \gamma_t^2 \beta^2}{\sigma_{\min}^2} E_t \\
&= E_t \Big( 1 - \mu \gamma_t + \frac{2\kappa \gamma_t^2 \beta^2}{\sigma_{\min}^2} \Big)
\end{aligned}
\tag{C.23}
$$

Restricting $\gamma_t$ such that

$$1 - \mu \gamma_t + \frac{2\kappa \gamma_t^2 \beta^2}{\sigma_{\min}^2} < 1 \iff \gamma_t < \frac{\sigma_{\min}^2 \mu}{2\kappa \beta^2}, \tag{C.24}$$

we finally have that, for the choice of $\gamma_t = \hat\gamma := \frac{\sigma_{\min}^2 \mu}{8\kappa \beta^2}$, it holds

$$E_t = \mathcal{O}(\rho^t) \quad \text{where } \rho := \Big( 1 - \frac{3\sigma_{\min}^2 \mu}{32\kappa \beta^2} \Big) < 1. \tag{C.25}$$

Leveraging the equivalence of $\mathrm{Err}_\Theta(\theta_t)$ and $\mathrm{Err}_\mathcal{X}(x_t) = E_t$ completes the proof. ∎

## D  Experiments

This section demonstrates the method's applicability in a series of different applccations. Along with revisiting the established examples of Section 5 in greater detail, we also present how the PHGD method performs in a couple of additional settings of interest. We begin with a high-level description of the common test suite, and afterward, we move to the definitive details of each of the presented applications.

In each example, we define a base game $\Gamma$ among two or more players $\mathcal{N} \equiv \{1, \dots, N\}$. Each player $i$ control a $m$-dimensional vector of control variables $\theta_i \in \mathbb{R}^m$, which they feed in an individual differentiable preconfigured MLP with two hidden layers that act as the player's representation map $\chi_i : \mathbb{R}^m \to \mathcal{X}_i$. The dimensions of each layer of the MLPs vary among the different examples. However, the MLP's output $x_i = \chi_i(\theta_i)$ is guaranteed to be a representation of a discrete probability distribution among the player's actions in some normal-form game, e.g., a Rock-Paper-Scissors game. The actual latent game $\mathcal{G}$ of $\Gamma$, is given by the "hidden" loss functions $f_i : \mathcal{X} \to \mathbb{R}$, $i \in \mathcal{N}$. Each loss function $f_i$ is a regularized variation of the aforementioned normal-form game, tuned by some $\frac{\mu}{2}$-strongly convex regularizer $h_\mu(x) = \frac{\mu}{2} \cdot \big( \sum_{i \in \mathcal{N}} \|x_i - x_i^*\|^2 \big)$, where $x^*$ is the normal-form game's unique equilibrium point. The modulus $\mu$ is a hyper-parameter, which we tune such that the vector field $g(x) = \big( \nabla_{x_i} f_i(x) \big)_{i \in \mathcal{N}}$ to be strongly monotone, and to avoid finite-precision numerical errors, which can potentially arise during the computation of $x$.

**A Hidden Game of Matching Pennies.**  As a first example, we revisit the two-player hidden game of Matching Pennies we introduced in Section 5. Here, each player's $i$ control variables $\theta_i$ are uni-dimensional, and are fed to each player's individual MLP given by the representation maps

$$\chi_i(\theta_i) = \mathrm{sigmoid}\big( \alpha_i^{(2)} \cdot CeLU(\alpha_i^{(1)} \cdot \theta_i) \big) \quad \text{for all } i = 1, 2, \tag{D.1}$$

where $\alpha_i^{(1)}, \alpha_i^{(2)} \in [-1, 1]$ are randomly chosen. Although the definition of the activation functions $\mathrm{sigmoid} : \mathbb{R} \to [0, 1]$, and $CeLU : \mathbb{R} \to (-1, \infty)$ are widely common, we give them below for reference; that is:

$$\mathrm{sigmoid}(x) = \big( 1 + \exp(-x) \big)^{-1} \tag{D.2a}$$

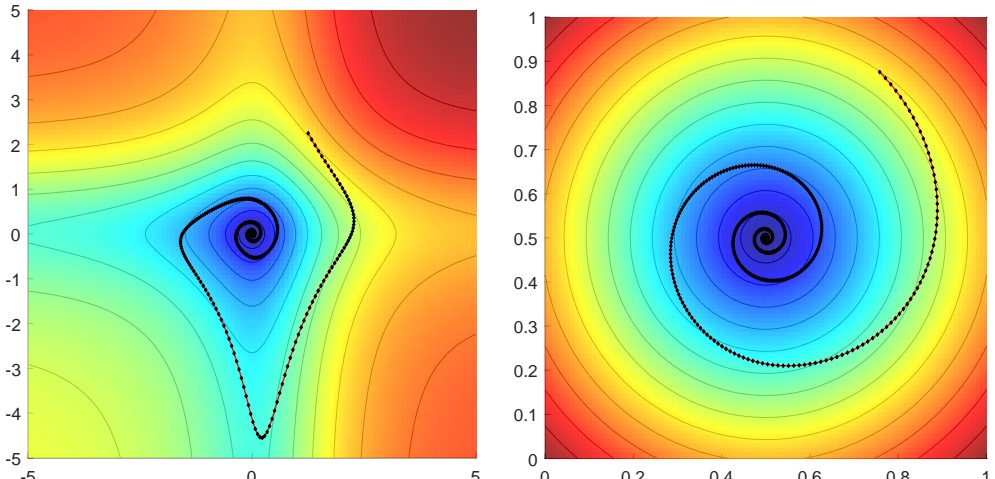

**(a)** A trajectory of PHGD in a regularized game of Matching Pennies over the sub-levels of the energy function in (5). The trajectory is depicted with reference, both, the space of control variables (left), and the space of latent variables (right). Notice that the trajectory crosses each sub-level set of the energy function, at most, once, indicating that the function's value is monotone along the trajectory.

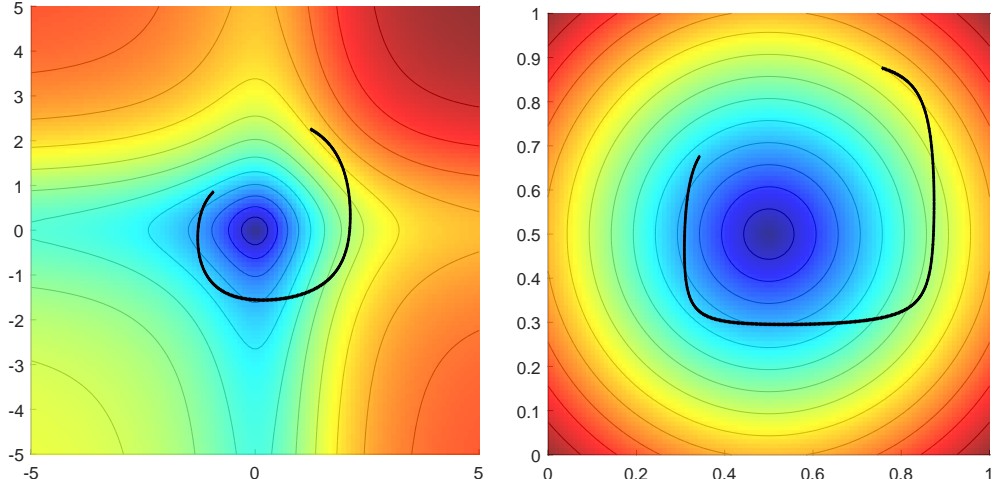

**(b)** A trajectory of GD in a regularized game of Matching Pennies over the sub-levels of the energy function in (5). The trajectory is depicted with reference, both, the space of control variables (left), and the space of latent variables (right). Notice that the trajectory crosses each sub-level set of the energy function multiple times indicating that the function's value is not monotone along the trajectory.

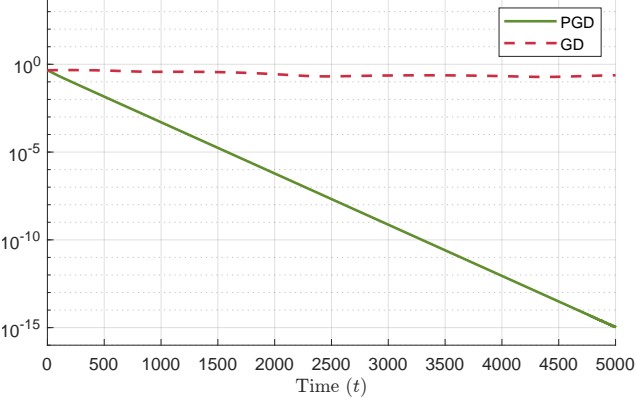

**(c)** The distance of $x$ to the latent game's equilibrium point $x^* = \left(\frac{1}{2}, \frac{1}{2}\right)$ along similar trajectories of PHGD and GD in a regularized game of Matching Pennies. The distance is depicted in a logarithmic scale in order to reveal the exponential rate of convergence of PHGD.

$$CeLU(x) = \max\{0, x\} + \min\{0, \exp(x) - 1\}. \tag{D.2b}$$

In this, and the following examples, without any loss of the generality, we deliberately set the bias of each of the MLP's hidden layers to zero, in order for the base game's unique equilibrium to lie in $\theta^* = \vec{0}$, and to simplify the notation. Each of the MLP's output $x_i = \chi(\theta_i)$ is the $i$-th player's probability of playing Heads in a regularized game of Matching Pennies, given by the loss functions

$$f_1(x) = -f_2(x) = -(2x_1 - 1) \cdot (2x_2 - 1) + h_{0.75}(x). \tag{D.3}$$

The latent game's unique equilibrium, in this case, is at $x^* = \chi(\theta^*) = (\frac{1}{2}, \frac{1}{2})$.

In this particular example it is possible to visualize the trajectories of PHGD and gradient descent with reference, both, the space of control variables $\mathbb{R}^2$, and the space of latent variables $[0, 1]^2$. In Fig. 7a we depict the trajectory of PHGD, with step-size 0.01, in the above game, initialized at the arbitrary state $(1.25, 2.25)$ with respect to, both, the space of control variables (left), and the space of latent variables (right), over the sub-level sets of the energy function in (5). A similar trajectory of the GD algorithm is depicted in Fig. 7b. Observe that the PHGD's crosses the sub-level sets of the energy function *at most once* until it asymptotically reaches the game's equilibrium point at $\theta^*$, as opposed to the trajectory of GD, which exhibits an erratic behavior. In particular, as is depicted in the semi-log plot in Fig. 7c, PHGD converges to the game's equilibrium point at an exponential rate as opposed to the rate of GD, which, at best, can be described as linear.

**A Hidden Game of Rock-Paper-Scissors.**   In the next example we consider a regularized hidden game of Rock-Paper-Scissors between two players. In a standard Rock-Paper-Scissors game each player may choose among three strategies, namely, Rock, Paper, or Scissors. No strategy dominates over the others, with Rock ruling over Scissors, Paper ruling over Rock, and Scissors ruling over Paper. In this example, each player $i$ controls a 5-dimensional vector $\theta_i \in \mathbb{R}^5$ of control variables, which, once again, they may feed their individual differentiable preconfigured MLP, whose output $x_i = \chi_i(\theta_i)$ lies in the 2-dimensional simplex that encodes the space of probability distributions among the three strategies of the latent game. The two MLPs are given by the following representation maps:

$$\chi_i(\theta_i) = \text{softmax}\left(A_i^{(2)} \times CeLU(A_i^{(1)} \times \theta_i)\right) \quad \text{for all } i = 1, 2, \tag{D.4}$$

where the matrices $A_i^{(1)} \in [-1, 1]^{4 \times 5}$, and $A_i^{(2)} \in [-1, 1]^{3 \times 4}$ are random, and the activation function $CeLU : \mathbb{R} \to (-1, \infty)$, given as in (D.2b), is applied pair-wise. The definition of $\text{softmax} : \mathbb{R}^d \to \Delta_{d-1}$, where $\Delta_{d-1}$ is the $(d-1)$-dimensional simplex is given, for reference, by

$$\text{softmax}_j(x) = \frac{\exp(x_j)}{\sum_{k=1}^d \exp(x_k)} \quad \text{for all } j = 1, \dots, d. \tag{D.5}$$

In this case, the latent game's loss functions are given by the following polynomial system of equations, whose unique equilibrium lies at the uniform distributions $x_i^* = \chi_i(\theta_i^*) = (\frac{1}{3}, \frac{1}{3}, \frac{1}{3})$, $i = 1, 2$:

$$f_1(x) = -f_2(x) = -x_1^\intercal A x_2 + h_{0.2}(x) \quad \text{where } A = \begin{bmatrix} 0 & -1 & 1 \\ 1 & 0 & -1 \\ -1 & 1 & 0 \end{bmatrix}. \tag{D.6}$$

Although, in this example, it is not possible to visualize the trajectories of PHGD and GD in the space of control variables due to the large dimensionality of the base game, we may still get a glimpse of the trajectories' behavior in the space of latent variables. Fig. 8a depicts an arbitrary trajectory of PHGD with step-size 0.01, as it evolves in the simplices of the two players. Notice, that the trajectory, clearly, converges to the equilibrium of the latent game. A trajectory of GD, with the same step size and initialization point, is depicted in Fig. 8b. In this case, the erratic behavior of GD is more apparent than in the previous example. A more in-depth comparison of the algorithms in the current setup is depicted in the semi-log plot of Fig. 8c, where we visualize the mean distance, and confidence bounds, to the equilibrium point across 100 random trajectories of the two algorithms. Observe, that PHGD exhibits an exponential rate of convergence, as opposed to the linear rate of GD.

**A Hidden Shapley's Game.**   The next game we are interested in is a hidden Shapley's game. The standard Shapley's game is a two-player normal-form game with payoff matrices:

$$A = \begin{bmatrix} 1 & 0 & \beta \\ \beta & 1 & 0 \\ 0 & \beta & 1 \end{bmatrix} \quad \text{and} \quad B = \begin{bmatrix} -\beta & 1 & 0 \\ 0 & -\beta & 1 \\ 1 & 0 & -\beta \end{bmatrix}, \tag{D.7}$$

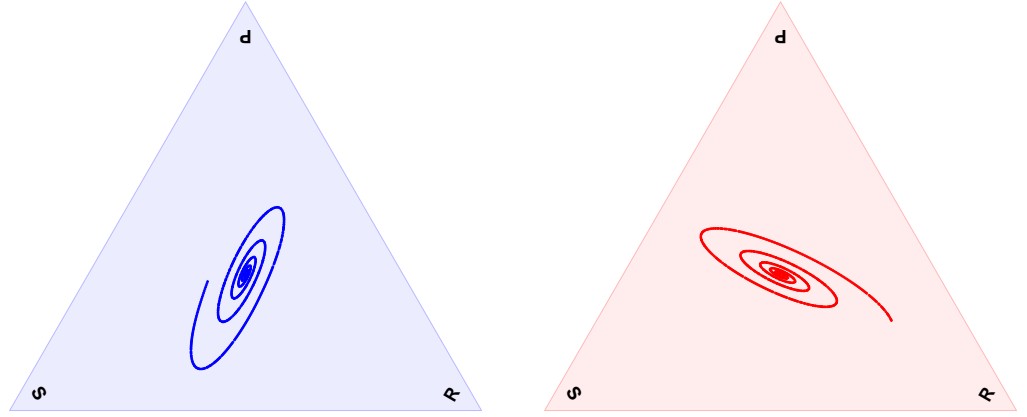

**(a)** A trajectory of PHGD in a regularized game of Rock-Paper-Scissors as depicted in each player's 2-dimensional simplex of latent variables. Notice that the trajectory converges to the latent game's equilibrium point $x^*$.

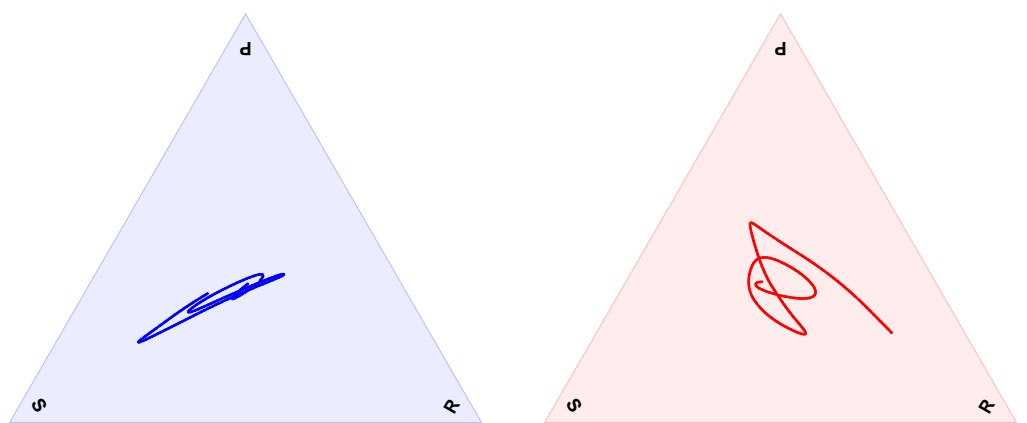

**(b)** A trajectory of GD in a regularized game of Rock-Paper-Scissors as depicted in each player's 2-dimensional simplex of latent variables. Notice that the trajectory exhibits erratic behavior.

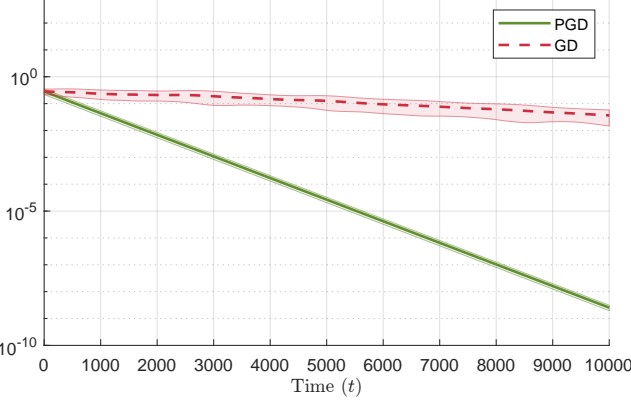

**(c)** The mean distance and confidence bound of $x$ to the latent game's equilibrium point $x^*$ along similar trajectories of PHGD and GD in a regularized game of Rock-Paper-Scissors. The distance is depicted in a logarithmic scale in order to reveal the exponential rate of convergence of PHGD.

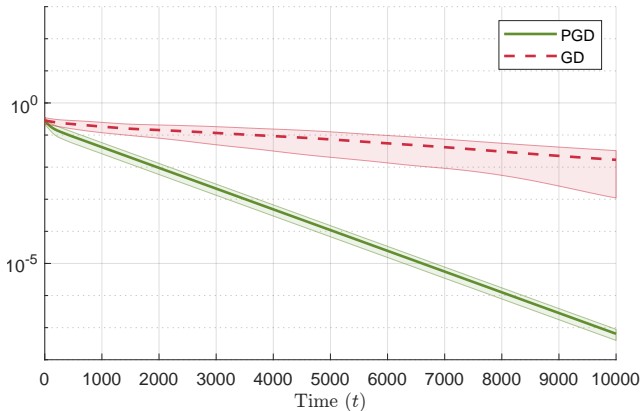

**Figure 9:** The mean distance and confidence bounds of $x$ to the latent game's equilibrium point $x^*$ along similar trajectories of PHGD and GD in a regularized Shapley game. The distance is depicted in a logarithmic scale in order to reveal the exponential rate of convergence of PHGD.

for some $\beta \in (0,1)$. The setup of this example is quite similar to the one in the previous example. However, there are a few noteworthy differences. To begin with, the hidden Shapley's game is not a zero-sum game. Specifically, the latent game's loss functions are given by the following polynomial system of equations:

$$f_1(x) = -x_1^\mathsf{T} A x_2 + h_{0.2}(x)$$
$$f_2(x) = -x_2^\mathsf{T} B^\mathsf{T} x_1 + h_{0.2}(x). \tag{D.8}$$

As in the hidden Rock-Paper-Scissors game, this game's unique equilibrium also lies in $x_i^* = \chi_i(\theta_i^*) = (\frac{1}{3}, \frac{1}{3}, \frac{1}{3})$, $i = 1, 2$. A second difference is the small modulus, $\mu = 0.2$, that we choose for the strongly-convex regularizer of this game. In Fig. 9 we depict a semi-log plot of the mean performance of PHGD and GD in the above game constructed using the same parameters as in the case of the hidden Rock-Paper-Scissors game. Notice that although the behaviour of PHGD is similar in both games, the confidence bounds of GD are drastically larger in the current example.

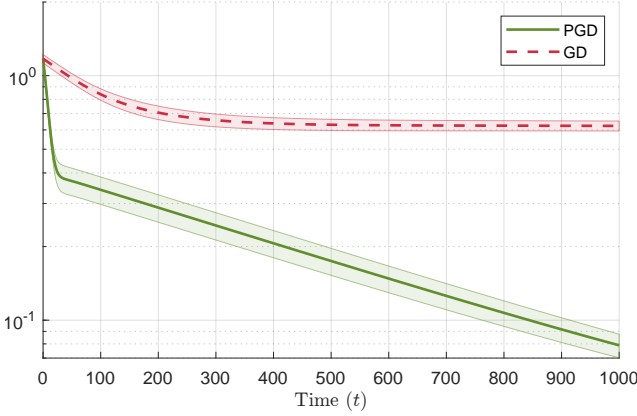

**Figure 10:** The mean distance and confidence bounds of $x$ to the latent game's equilibrium point $x^*$ along similar trajectories of PHGD and GD in a regularized El Farol Bar game. The distance is depicted in a logarithmic scale in order to reveal the exponential rate of convergence of PHGD.

**A Hidden El Farol Bar Game.** As a final example, we are going to revisit and describe the details of the hidden El Farol Bar game we introduced in Section 5. The El Farol Bar game is a famous congestion game that is often described as a game between populations. However, in this example, we are interested in its atomic $N$-playe variant. In the standard El Farol Bar game each player is given the option of visiting a specific bar in El Farol, and the outcome of the game is determined based on the number of players that decided to visit the bar. From the perspective of the player, there are three possible situations they may encounter, which carry some respective payoff. If the player

decides to do not to visit the bar, then, independently of the number of bar tenants, the player receives a payoff $S$. On the other hand, if the player decides to visit the El Farol bar, then depending on how crowded the bar is, they may receive a payoff that is strictly smaller, or strictly larger than $S$. If the bar is crowded, i.e., more than $C \geq 0$ other players have visited the bar at the same time, then each of them receives payoff $B < S$. However, if the bar is not crowded, i.e., the total number of tenants is less than $C$, then they receive payoff $G > S$. It's not difficult to verify that the El Farol Bar game has a unique mixed Nash equilibrium, where each player chooses to visit the bar with probability $\frac{C}{N}$.

We are going to consider a hidden variant of the above game among $N = 30$ players. Each player $i$ controls a $5$-dimensional vector of control variables $\theta_i \in \mathbb{R}^5$, which feeds to an individual, differentiable, and preconfigured MLP. The MLP's output $x_i = \chi_i(\theta_i)$ is guaranteed to lie in $[0, 1]$ and is interpreted as the probability of player $i$ visiting the El Farol bar. Regarding the MLP configuration, we follow a similar structure as in the previous examples. Specifically, the representation maps $\chi_i : \mathbb{R}^5 \to [0, 1]$ are defined as

$$\chi_i(\theta_i) = \text{sigmoid}\big(\alpha_i^{(2)} \cdot CeLU(A_i^{(1)} \times \theta_i)\big) \quad \text{for all } i = 1, 2, \tag{D.9}$$

where $A_i^{(1)} \in [-0.85, 0.85]^{4 \times 5}$, and $\alpha_i^{(2)} \in [-1, 1]^4$ are randomly chosen. The additional restriction to the domain of $A^{(1)}$ reduces the chance for over-flow numerical errors of the sigmoid activation function. The activation function $CeLU : \mathbb{R} \to (-1, \infty)$ is given as in (D.2b) and is applied pair-wise to the output of $A_i^{(1)} \times \theta_i$. The loss functions of this game follow the standard variant's definitions, namely, we have that

$$f_i(x) = S + x_i \Big( G - S + \mathbb{P}\Big(\sum_{i \neq j} x_i \geq C\Big)(B - G) \Big) + h_{0.5}(x) \quad \text{for all } i \in \mathcal{N}, \tag{D.10}$$

and the latent game's unique equilibrium is at $x_i^* = \frac{C}{N}$, $i \in \mathcal{N}$.

The large dimensionality of the above game prohibits the usage of a detailed visualization as opposed to the previous examples. However, sufficient information about the behavior PHGD, and GD can be gathered by the performance log-plot across 100 random trajectories in Fig. 10. Observe that, regardless of the increased number of players, the behavior of PHGD is unaffected, i.e., it converges to the game's equilibrium at an exponential rate. On the other hand, the GD fails to converge in the game's equilibrium. In fact it eventually maintains a constant distance from it, unable to procceed further; an indication of a cycling behavior.

