# OpenReview forum: "Exploiting hidden structures in non-convex games for convergence to Nash equilibrium"
_NeurIPS.cc/2023/Conference — NeurIPS 2023 poster_

### Official Review · Reviewer_zCUr · 2023-06-28

**Soundness:** 3 good
**Presentation:** 2 fair
**Contribution:** 3 good
**Rating:** 6
**Confidence:** 2

**Summary:**

This paper proposes a preconditioned hidden gradient descent to provide strong formal convergence guarantees in a general class
of multi-agent settings which are referred to as call hidden monotone games. Theoretical analyses and synthetic experiments are also provided.

**Strengths:**

1. The method seems novel to handle the proposed problem of hidden monotone games.
2. Theoretical analyses and synthetic experiments are also provided to demonstrate the effectiveness of the proposed method.

**Weaknesses:**

1. Experiments on real-world datasets (e.g., MNIST or multi-agent reinforcement learning environments) are absent.
2. Experiments on adversarial generation, adversarial attack, and adversarial transfer learning are absent.
3. Computational complexity analyses are absent.

**Questions:**

1. Is it feasible for this method to be applied to common neural networks, such as ResNet50, BERT, etc?

**Limitations:**

Yes.

---

> ### Author Rebuttal · Authors · 2023-08-10
>
> Thank you for your input and positive evaluation. We reply to your questions point-by-point below:
>
> 1. **Discussion about the experiments included in the paper:**
> Our work distinctly stands out for its depth and novelty in experimental design within the realm of hidden games. A review of recent studies, such as [1,2,3], showcases their focus on continuous dynamics, offering only a fleeting glance at experiments with discrete step-sizes. Specifically:
> - Their experiments, although illustrative, were limited to synthetic setups, often featuring a single hidden variable and straightforward mappings, like the invertible sigmoid in games such as Rock Paper Scissor.
> - [3] ventured slightly further, introducing a single hidden layer, but fell short in detailing the nuances of converting continuous dynamics to a discrete format using the Euler method.
>
> In stark contrast, our experiments probe deeper, navigating games with multiple hidden layers using our distinctive method.
> Furthermore, our appendix introduces a seminal benchmark for Nash computation in expansive policy domains, exemplified by the El-Farol Bar (see [4] and references therein).
>
>
> 2. **Discussion about real-world application, MARL or adversarial training:**
>
>     Our primary objective was to devise a game theoretically applicable algorithm for computing Nash equilibria in structured and ML-inspired non-convex settings. Unlike prior works, ours introduces theory and examples with intricate hidden maps.
>
>     Training MARL and the other applications are outside the scope of the current paper. However, the strength of our theoretical analysis distinctly underscores its applicability. In particular, when evaluating works such as [5] and [6] (highlighted by another reviewer for a comparative examination of related work), our framework and algorithm not only address challenges in variational policy methods within MARL and the econometric literature but do so with a broader range of assumptions than those outlined in [5,6].
>
>     Thus, through establishing the theoretical basis for PHGD, we anticipate inspiring engineers and practitioners to adapt PHGD to large-scale challenges.
>
>
> 3. **On the computational complexity of Hidden Games:**
>
>    The computational complexity of Hidden Games is an open and captivating topic. First, it's essential to focus on loss functions/utilities and hidden maps with a succinct polynomial representation under certain well-behaved arithmetic circuits [7]. Without this, defining a meaningful complexity result inside NP for continuous strategy games becomes challenging. When arithmetic circuits that describe losses are generic, the problem becomes FIXP-hard [15]. When equipped with these succinct models, we can efficiently compute value, gradient, and Jacobian oracles for hidden maps in polynomial time. Within this modeling framework, the problem could be shown to be at least as hard as PPAD (the standard complexity class for computing ε-Nash equilibrium ).  On the other hand, proving membership to PPAD for general hidden maps — without escalating higher in the computational hierarchy of TFNP — remains an open question. While the authors of [8] outlined a meta-method using a version of Kakutani's theorem, the specifics are yet to be fleshed out.
>
>    *All this underscores why the assumptions we stipulated are vital for a positive outcome in computing Nash equilibrium in non-concave general games.* Furthermore, when more structures like diagonal convexity/monotonicity are introduced, our results provide an algorithm for finding the Nash equilibrium. **Using our algorithm, PHGD, and given certain value/gradient oracle access, the problem is actually determined to lie in P based on iteration complexity results.**
>
>    The discussion on computational complexity doesn't end here. If we assume only black-box access to the objective, [9-11] derive unconditional lower bounds for the general hidden games. However, a more detailed account goes well beyond the scope of the current work, so it is not possible to go into more length here.
>
>
>
> 4. **Regarding the utilization of more general neural networks like BERT/ResNet:**
>
>     In essence, the crux of our theoretical requirements can be distilled to: *if the neural networks are highly expressive, our method can reliably uncover the NE in associated games.* The empirical evidence surrounding the aforementioned networks suggests they meet the prerequisites our theorems mandate. Developing a theoretical framework for such Deep Learning Game Theory would likely necessitate an average/smoothed complexity analysis. Recent findings related to learning models with NNs [12,16] and computing Nash equilibria in generalized random games [14] might provide an intriguing trajectory for subsequent research.
>
>     Given the potential interest in this topic within the community, we will aim to incorporate elements of this discussion into the appendix or the camera-ready version, focusing on future and open challenges.
>
> [1] https://arxiv.org/abs/1910.13010
> [2] https://arxiv.org/abs/2101.05248
> [3] https://openreview.net/forum?id=bsycpMi00R1
> [4] https://arxiv.org/abs/2106.01285
> [5] https://arxiv.org/abs/2007.02151
> [6] https://arxiv.org/abs/2205.01774
> [7] https://arxiv.org/abs/2011.01929
> [8] https://arxiv.org/abs/2207.07557
> [9] https://arxiv.org/abs/2009.09623
> [10] Exponential lower bounds for finding Brouwer fixed points
> [11] Problem complexity and method efficiency in optimization
> [12] https://arxiv.org/abs/2302.07426
> [13] Book Computational Complexity.
> [14] https://arxiv.org/abs/2007.10857
> [15] https://arxiv.org/abs/2111.06878
> [16] https://arxiv.org/abs/2211.03975

---

### Official Review · Reviewer_rghq · 2023-07-04

**Soundness:** 3 good
**Presentation:** 3 good
**Contribution:** 4 excellent
**Rating:** 7
**Confidence:** 3

**Summary:**

It is known that a convergence guarantee exists for monotone games. However, most games are not monotone. This paper considers a new scenario where the monotone structure is presented in latent space. It then proposes the 'Preconditioned Hidden Gradient Dynamics' to design the preconditioned hidden gradient descent algorithm. It is demonstrated that the proposed algorithm will converge to the desired Nash equilibrium. The theoretical result is also verified using empirical experiments.

**Strengths:**

Novelty: Existing works have shown the difficulties of finding the equilibrium in hidden-monotone games. Other results usually require additional assumptions on the structure. This work proposes a new perspective to design the preconditioned hidden gradient descent and solves the problem that has not been addressed before.

Significance: Most of games are not monotone. Studying the hidden-monotone structure will significantly extend the existing literatures on monotone games, which makes me believe this work opens a new direction of the resarch. The positive result obtained in this work and the new design in the continous-flow will also help understanding on the hidden structure.


**Weaknesses:**

This paper is very well-written. But it may lack some real-world examples to help the reader understand the importance of studying the hidden games.

**Questions:**

This paper has mentioned many times "minimal assumptions" or "without additional assumptions". I am probably interested in how strong other assumptions are, such as separability or low-dimensionality assumptions. Do you have any simple examples saying these assumptions are not satisfied.

Figure 1 is very helpful to understand the concept of hidden games. Do you have any practical example or real-world example that would have such Rock-Paper-Scissors hidden structure?

**Limitations:**

This is a purely theoretical work so no negative impact.

---

> ### Author Rebuttal · Authors · 2023-08-09
>
> Thank you for your support and your helpful comments.
>
> 1. **This paper is very well-written. But it may lack some real-world examples to help the reader understand the importance of studying the hidden games.**
>
>     We are glad to hear that you found our paper very well written! A natural example would be to focus on the competition between two companies that fight for market share. Specifically, let's consider the competition between two ride-share platforms such as Uber vs Lyft. Recent work has produced explicit models of such competition under the assumption that the demand for each company at each location depends linearly on the set of posted prices by each company [1]. Furthermore, they show that these models correspond to a strongly monotone game and then apply dynamics to solve them. Clearly, the linear demand model is a simplified assumption. By introducing a non-linear model (that is linear only on some space of latent variables) we can capture much more complex, real-world dependencies on the elasticity of demands while satisfying hidden monotonicity.
>
>
> [1] Narang, Adhyyan, et al. "Learning in stochastic monotone games with decision-dependent data." International Conference on Artificial Intelligence and Statistics. PMLR, 2022.
>
> 2. **This paper has mentioned many times "minimal assumptions" or "without additional assumptions". I am probably interested in how strong other assumptions are, such as separability or low-dimensionality assumptions. Do you have any simple examples saying these assumptions are not satisfied.**
>
>     Sure thing, consider e.g., the weighted softmax function $\chi(\theta_1,\theta_2) = \log\left[(1+\theta_2^2)e^{\theta_1} + (1+\theta_1^2)e^{\theta_2}\right]$. This is neither one-dimensional nor separable.
>
>
> 3. **Figure 1 is very helpful to understand the concept of hidden games. Do you have any practical example or real-world example that would have such Rock-Paper-Scissors hidden structure?**
>
>     We kindly refer you to our answer to the first point of our discussion, as this is a special case.

---

> > ### Comment · Reviewer_rghq · 2023-08-14
> > **Response**
> >
> > Thank you for the detailed examples! My concerns have been well addressed. I've also gone through comments from other reviewers, but none raised any new concerns for me. So I will maintain my positive rating.

---

### Official Review · Reviewer_dBPZ · 2023-07-06

**Soundness:** 3 good
**Presentation:** 3 good
**Contribution:** 4 excellent
**Rating:** 7
**Confidence:** 4

**Summary:**

The paper uses hidden structure to provide continuous time and algorithmic theoretical learning guarantees for certain non-convex games. Specifically, they provide Preconditioned Hidden Gradient Flow and its discrete-time variant Preconditioned Hidden Gradient Descent that can be proved to converge when analyzed under a specific Lyapunov function.


**Strengths:**

The theoretical contribution of the paper is significant, compared to past work, the setting is significantly more general. The paper is very well presented, with figures explaining the basic theoretical ideas behind the paper. The theory is solid and rigorous, while I have not checked the proofs line by line, they seem to be correct. Furthermore, significant motivation is provided both visually and by argumentation for the proof strategies employed.

Experiments are also provided, and in toy examples the algorithm is demonstrably better than alternatives. Overall, the paper solves an important problem in algorithmic game theory.

**Weaknesses:**

I think this is a strong paper with a solid contribution. Some possible weaknesses:

* A discussion of the relevance of the faithfulness assumption on the hidden map could be discussed. For instance, the experiments conducted seem to be on particular toy problems, it’s not immediately clear if there are large classes of problems with hidden structure that satisfy the given conditions.
* The authors make an effort to present the ideas behind the proofs, which are the main contributions of the paper. Still, the presentation in the main body could benefit from a condensed roadmap of the proof strategy. While Section 4 contains main ideas, the overall picture was difficult to grasp.
* The assumption on the representation map (e.g. faithfulness) is difficult to locate, the assumption could be made more explicit for ease of reading.
* Experiments are limited to somewhat simple cases, an evaluation of PHGD on for instance GAN training or other relevant problems in practice could be a stronger empirical argument.

**Questions:**

The examples mentioned in the experiment are interesting but also seem to be restricted to very specific cases. Are there larger classes of problems besides Examples 2.1, 2.2 that could be relevant in applications? For instance, has there been work on when neural network parameterization could satisfy the faithful map condition? While being possibly out of the scope of the main contribution of the paper, a discussion of these points could be interesting.

As a possible limitation, is the implied access to the Jacobian of representation map standard in the literature?

The paper mentions dimension reduction as a motivation of the representation map. It is not immediate to me that dimension reduction could be achieved while simultaneously preserving the faithfulness assumptions, due to topological considerations. A discussion of this point could be interesting. For instance, does the El Farol Bar game parameterization satisfy the faithfulness assumptions?

*Minor questions:*

Line 195 -> Is the message that $\mathbf{P}_i$ will be designed to achieve certain properties of was it defined before? Was not clear on the first read.

Lines 177 - 189 -> The connection between the two paragraphs was not clear. Was NHGD shown to not require the mentioned assumptions?

*Possible minor Typos:*
Line 116: $\Theta$ -> $\Theta_i$?
Line 164: For the Jacobian notation, should it depend on $\theta_i$
Line 172: Here, it is possible that the notation is confusing with the previous separation of the control variables of each player. As far as I understand, the individual coordinates of the controls of one player is meant here.
Line 254: an closed -> a closed

**Limitations:**

No major limitations. As mentioned above: empirical validation could be extended to relevant problems in application.

---

> ### Author Rebuttal · Authors · 2023-08-10
>
> Thank you for your encouraging remarks and your positive evaluation. We reply to your questions point-by-point below:
>
> 1. **A discussion of the relevance of the faithfulness assumption on the hidden map could be discussed.**
>
>     We will be happy to add a discussion, however, we believe that it could be more useful to see our setting as more prescriptive than descriptive - in the sense that we provide an intuitive set of conditions that allow for strong theoretical guarantees. Our contributions can help the design of practical systems with better performance and stability.
>
>
> 2. **The presentation in the main body could benefit from a condensed roadmap of the proof strategy.**
>
>     Sure thing, if accepted, we would be happy to take advantage of the extra page allowed in the revision to include such a roadmap. Thank you for the suggestion!
>
>
> 3. **The assumption on the representation map (e.g. faithfulness) is difficult to locate, the assumption could be made more explicit for ease of reading.**
>
>     Of course, we will be happy to collect it with the rest of the assumptions in the revision stage.
>
>
> 4. **An evaluation of PHGD on GAN training or other relevant problems in practice could be a stronger empirical argument.**
>
>     Although GAN training is undisputably an important problem, we would like to point out that in game-theoretic setup it corresponds to a 2-player Hidden Convex-Concave Game. Some preliminary experimental results in this setup can be traced to the work of Mladenovic et al. GAN training however is a 2-player game, while in this paper we'd like to provide a method that goes beyond the 2-dimensional barrier. The examples we present build toward that goal, starting with examples that provide good intuition and visual feedback, and eventually arriving at the El Farol Bar game, a hard $n$-player benchmark.
>
>
> 5. **Are there larger classes of problems besides Examples 2.1, 2.2 that could be relevant in applications?**
>
>     We kindly refer the reviewer to Appendix D where we provide empirical evidence in a variety of examples, including a hidden El Farol Bar structure, which is a highly non-trivial $n$-player game. Furthermore, we would like to point out that the complexity of the type of problems we study arises from at least two factors, namely, the complexity of the base game, and the complexity of the associated latent game. For example, in a Deep Learning setup, while the complexity of the base game played in the input layer of the MLPs is high, the latent game, played to, or near, the output layers of the MLPs is comparatively lower. In the examples we provide in Section 5 and Appendix D we provide evidence of the applicability of PHGD in setups where the complexity of the base game is high relative to the capabilities of our own equipment, but also the associated latent games are gradually increasing in difficulty in each of the examples.
>
>
> 6. **Has there been work on when neural network parameterization could satisfy the faithful map condition?**
>
>     We kindly refer you to our answer to the first point of our discussion.
>
>
> 7. **As a possible limitation, is the implied access to the Jacobian of representation map standard in the literature?**
>
>     Yes, indeed: in the literature on hidden games, the latent structure - that is, the map $\chi$ or its equivalent - is assumed known, so the calculation of the Jacobian is also accessible by the same tenet.
>
>
> 8. **The paper mentions dimension reduction as a motivation of the representation map. It is not immediate to me that dimension reduction could be achieved while simultaneously preserving the faithfulness assumptions, due to topological considerations. A discussion of this point could be interesting. For instance, does the El Farol Bar game parameterization satisfy the faithfulness assumptions?**
>
>     Any linear map $(A:\mathbb{R}^m \to \mathbb{R}^d)$ with full column rank is faithful so, if $m>d$, generic linear maps are faithful. Then, taking a composition with a faithful activation function (for instance, a leaky ReLU) would lead to a faithful representation, so there are no topological obstacles in this regard. [In our "faithful" does not mean injective / "1-1" in our context; this would indeed be topologically impossible.]
>
>
> 9. **Line 195 -> Is the message that $P_i$ will be designed to achieve certain properties or was it defined before? Was not clear on the first read.**
>
>     The idea was that $\mathbf{P}$ would be designed to satisfy certain key properties; apologies if this was not clear, we will clarify this in the revision round.
>
>
> 10. **Lines 177 - 189 -> The connection between the two paragraphs was not clear. Was NHGD shown to not require the mentioned assumptions?**
>
>     In our discussion, the work of Mladenovic is, in a sense, "sitting on the fence". To elaborate, they sidestep some of the more restrictive assumptions articulated in the introductory paragraph of Section 3. Notably, their portrayal of NHGD eschews any necessity for a one-dimensional representation, instead honing in on the latent space rather than the parameter domain.
>
>     To recast this in our notation for clarity: Mladenovic et al. refrain from prescribing a distinct structure to the latent maps, denoted as $\chi_i(\theta_i)$ for $i=1,2$. This approach marks a progression from the strategies adopted by Vlatakis et al. However, it remains imperative to recognize the confines of Mladenovic et al.'s methodology. Among various limitations, their model is primarily tailored for a 2-player scenario. Additionally, upon examining their convergence justification (as delineated in Proposition 4.1 of their paper) and subsequent demonstrations regarding NHGD, their analysis only applies to one-dimensional problems.
>
>
> 11. **{Minor typos}**
>
>     Will fix them, thanks for bringing them to our attention!
>
> ---
>
> Thank you again for your input and positive evaluation - and please let us know if you have any further questions.

---

> > ### Comment · Reviewer_dBPZ · 2023-08-11
> >
> > Thanks for the clarifications.
> >
> > Regarding the matter of dimension reduction: Assumption 1 still requires that the Jacobian has singular values lower bounded away from zero. By the inverse function theorem then we can find (at least) a neighborhood where the representation map is locally continuous bijection to the image of the neighborhood, implying an open neighborhood of the parameter space and the latent space is homeomorphic and has the same dimension. In fact, in the example given by the authors, the linear map would have non-empty nullspace (implying a 0 singular value for the Jacobian). Or am I missing something?

---

> > > ### Author Response · Authors · 2023-08-12
> > >
> > > Thanks for your follow-up, we understand now the source of the confusion!
> > >
> > > To make things absolutely clear, consider first as an example the linear transformation $J\colon\mathbb{R}^2 \to \mathbb{R}$ with matrix
> > > $$
> > > J = \begin{matrix}(1 & 2)\end{matrix}
> > > $$
> > > The singular values of $J$ are defined either as the eigenvalues of $J^\top J$ or as the eigenvalues of $JJ^\top$, depending on convention. In the former convention, the singular values of $J$ are $\sqrt{5}$ and $0$, in the latter there is only one, $\sqrt{5}$. The latter convention is more parsimonious for linear transformations that are surjective, so this is the convention we tacitly employed throughout (precisely because the regime of interest is when $m \geq d$). [When there is danger of confusion, the latter convention is sometimes referred to as "thin singular value decomposition" in the literature]
> > >
> > > With this in mind, the relevant part of Assumption 1 reads more explicitly as
> > > $$
> > > \sigma_{\min}^2 \leq \mathrm{eig}(J_\theta J_\theta^\top) \leq \sigma_{\max}^2
> > > $$
> > > so there is no issue if $m>d$: in this case, $J_\theta \colon \mathbb{R}^m \to \mathbb{R}^d$ is surjective (or, equivalently, $J_\theta$ has full rank as a matrix). In particular, the inverse function theorem *does not apply* for $m>d$ (since $J_\theta$ cannot be injective if it is surjective and $m>d$), but the *implicit* function theorem does, and it implies that $\chi$ locally looks like a projection (in suitable coordinates around the point under study).
> > >
> > > We hope this clears things up. To avoid any risk of ambiguity or confusion and make things more explicit, we will restate Assumption 1 directly in terms of the eigenvalues of $J_\theta J_\theta^\top$.
> > >
> > > Thank you again for the follow-up - and please let us know if you have any further questions!
> > >
> > > Kind regards,
> > >
> > > The authors

---

> > > > ### Comment · Reviewer_dBPZ · 2023-08-14
> > > >
> > > > Thank you very much for the clarification and the explicit statement of Assumption 1.

---

### Official Review · Reviewer_hHrj · 2023-07-09

**Soundness:** 3 good
**Presentation:** 2 fair
**Contribution:** 4 excellent
**Rating:** 5
**Confidence:** 3

**Summary:**

The paper focuses on studying non-convex games with hidden structures, where latent variables can be seen as a function of control variables and are decoupled. The authors propose a discrete algorithm called Preconditioned Hidden Gradient Descent (PHGD) to exploit the hidden structure and achieve convergence to Nash equilibrium. The method establishes a connection with natural gradient methods through the selection of gradient preconditioning schemes. The paper provides the first discrete convergence analysis on hidden convex concave games and extends the separable assumption to a more general multi-player setting. Convergence is proven under minimal assumptions, both in deterministic and stochastic environments.

**Strengths:**

- Originality: The paper introduces a novel algorithm, PHGD, to address non-convex games with hidden structure, providing the first discrete convergence analysis in this context. The connection established with natural gradient methods adds an original perspective to the research.

- Quality: The paper demonstrates rigorous analysis by providing explicit convergence rates under minimal assumptions. It extends the previous work by considering a more general multi-player setting.

- Clarity: The summary provided is clear and understandable, indicating good clarity in the paper.

- Significance: The paper addresses an important problem by leveraging hidden structure in non-convex games, which has implications in various domains. The developed algorithm and its convergence properties contribute to the understanding and applicability of game theory.

**Weaknesses:**

- Lack of diverse examples: The paper could benefit from including additional examples beyond the matching Pennis zero-sum game to illustrate the applicability of the proposed method in different settings. The example shown in this paper can be included within previous 1-dimensional assumptions presented Mladenovic et al. Are there significant instances that this paper covers while previous results do not?

- Lack of comparison with existing methods: It would be helpful if the authors provided a more detailed explanation of the technical challenges associated with discrete analysis compared to continuous ones. For the continuous dynamics, does it reduces to the exact natural gradient flow in hidden convex-concave games in Mladenovic et al.,

- It seems that the faster convergence is at the expense of high computational complexity.

**Questions:**

- Preconditioning method and its relation to natural gradient: The authors claim that "a judiciously chosen gradient preconditioning scheme bears an unexpected connection to natural gradient method". However, the connection has not been well explained in the main context. What is the literature of the preconditioning method? Why is the previous natural gradient approach not seen as preconditioning?


- Technical difference in extending to the multi-dimensional case: What are the main technical differences or challenges encountered when extending the approach to the multi-dimensional case?

- Minor issue regarding notation: In Line 195, where $P_i(\theta_i)$ appears before introducing what $P_i$ represents.

**Limitations:**

The limitations of this paper have not been addressed adequately.

---

> ### Author Rebuttal · Authors · 2023-08-09
>
> Thank you for your input and detailed remarks. We address each of your questions point-by-point below and we will revise our manuscript accordingly at the first revision opportunity.
>
> 1. **Lack of diverse examples: The paper could benefit from including additional examples beyond the matching Pennies zero-sum game to illustrate the applicability of the proposed method in different settings.**
>
>     We kindly refer you to Appendix D where we demonstrate the algorithm's applicability in a series of different examples.
>
>
> 2. **The example shown in this paper can be included within previous 1-dimensional assumptions presented Mladenovic et al. Are there significant instances that this paper covers while previous results do not?**
>
>     The focus of Mladenovic et al (ICLR 2021) is a continuous-time setup. In our work, although we begin by developing a more general continuous-time method than that of Mladenovic et al (Section 3.1), we continue by developing a bona fide, discrete-time algorithm; more importantly, we then go on to provide theoretical guarantees about its convergence, and its rate of convergence (Section 3.2 onwards). This is a crucial distinction from Mladenovic et al, who do not provide an algorithmic analysis of the proposed dynamics.
>
>
> 3. **Lack of comparison with existing methods: It would be helpful if the authors provided a more detailed explanation of the technical challenges associated with discrete analysis compared to continuous ones.**
>
>     We should first point out that, to the best of our knowledge, all dynamics that have been proposed for solving hidden games evolve in *continuous* time; any iterative algorithm that can be programmed in a computer would have to de facto evolve in *discrete* time, so there are no other *algorithms* to compare to.
>
>    Now, as for the difficulty of going from continuous to discrete time, this has been the driving force for the development of stochastic approximation theory, starting from the original works of Robbins & Monro in the 1950s to the classical textbooks of Kushner and co-authors in the 1970s, and Benaïm in the 1990s. To make an extremely long story extremely short, when performing a Lyapunov/energy analysis of a continuous-time system, the chain rule of ordinary calculus suffices; however, when the analysis needs to be performed at discrete time steps, the step size plays a crucial role, as it introduces further error terms that propagate and require completely different techniques to handle. Then, when noise is also present in the mix, the situation becomes even more complicated because one needs to employ martingale limit theory to control the various stochastic errors that arise - and which are completely absent in the continuous-time flow limit (which is de facto deterministic).
>
>
> 4. **For the continuous dynamics, does it reduces to the exact natural gradient flow in hidden convex-concave games in Mladenovic et al.?**
>
>     The dynamics featured in Mladenovic et al can indeed be viewed as a special case of Preconditioned Hidden Gradient Flow. However, note that Mladenovic et al feature the aforemntioned formula for the case of a single datapoint (i.e., one-dimensional systems). Their general formula is not directly comparable.
>
>
> 5. **It seems that the faster convergence is at the expense of high computational complexity.**
>
>     We are not sure what you mean here: the per-iteration complexity of the algorithm remains the same throughout, so a faster convergence rate would be synonymous with lower computational complexity. Could we kindly ask you to elaborate if needed?
>
>
> 6. **Preconditioning method and its relation to natural gradient: The authors claim that "a judiciously chosen gradient preconditioning scheme bears an unexpected connection to natural gradient method". However, the connection has not been well explained in the main context. What is the literature of the preconditioning method? Why is the previous natural gradient approach not seen as preconditioning?**
>
>     Preconditioning simply means finetuning the search direction of an iterative optimization algorithm to make more informed gradient steps. It is a technique that permeates the optimization literature and is the cornerstone of some of the most widely used methods - like L-BFGS, CG, and the like; see e.g., the classical textbook of Himmelblau, "*Applied Nonlinear Programming*".
>
>     Natural gradient methods essentially redefine the gradient operator, so they can be seen as a special case of preconditioning. What we found unexpected was the fact that the choice of preconditioner which was the most appropriate for our analysis ended up being itself a natural gradient method (and included the continuous-time method of Mladinovic et al.). We were constrained by space in our original submission but, if the paper is accepted, we will use the extra page allowed in the camera-ready phase to explain all this in more detail.
>
>
> 7. **Technical difference in extending to the multi-dimensional case: What are the main technical differences or challenges encountered when extending the approach to the multi-dimensional case?**
>
>     The original approach of Mladinovic et al (ICLR 2021) is inherently one-dimensional and requires separability: if either condition fails, we see no way of extending it. In this regard, the main technical difficulty was in guessing the form of the preconditioner that would allow us to carry out a Lyapunov analysis.
>
>
> 8. **Minor issue regarding notation: In Line 195, where $P_i(\theta_i)$ appears before introducing what represents $P_i$.**
>
>     Good catch, we will add a reference to equation PHGF on that line.
>
>
> ---
>
> Please let us know if any of the above is not sufficiently clear.

---

> > ### Comment · Reviewer_hHrj · 2023-08-18
> >
> > Thank you for your rebuttal. I believe the contribution of this work is good, but there is still room for improvement in its presentation. I will raise my score to 5.

---

> > > ### Author Response · Authors · 2023-08-18
> > >
> > > Thank you for your positive re-evaluation, we will update our paper according to your detailed input and remarks.
> > >
> > > Regards,
> > >
> > > The authors

---

### Official Review · Reviewer_cb9a · 2023-07-31

**Soundness:** 2 fair
**Presentation:** 1 poor
**Contribution:** 3 good
**Rating:** 5
**Confidence:** 4

**Summary:**

This paper studies a hidden preconditioned stochastic gradient descent method for finding Nash equilibrium in games with hidden monotone structures. It demonstrates non-asymptotic convergence bound of the proposed algorithm. The complexity bound of the hidden monotone game matches that of monotone games. The topic is very interesting as it provides a way to handle non-monotone stochastic games, which motivates me to read in details. Overall, it is an interesting paper. However, the presentation quality needs serious improvement. It is hard to judge the correctness with so many typos.

**Strengths:**

The motivation of the hidden gradient descent design is clear.

The convergence rate result for hidden monotone stochastic games seems to be new and matches the optimal complexity bound for stochastic games.




**Weaknesses:**

The presentation of the paper is poor with lots of typos. Without the Errata provided by the authors in the Appendix, when reading the main context for the first time, I found many mathematical mistakes. After reading the authors own corrections, there still remain many typos, making it hard to understand. The notation system is very confusing. I have tried my best to go through the proof. The overall idea seems correct but I cannot guarantee details. See more comments below.

The examples used to motivate the problems seems to be too simple. Example 2.1 is too simple. Example 2.2, it is unclear why one does not first solve the bilinear problem and find a corresponding mapping, which is not even needed actually. The example provided in figure 1 does not really need a neural network to approximate the probability space.

For Assumption 1, the paper claims that $\theta$ and $x$ does not need to be within the same space, i.e., the dimension of them can be different. I wonder if $\Theta$ is the whole space on $R^m$ and if $\mathcal{X}$ is the whole space on $R^d$. Please specify the conditions for the Jacobian has positive and finite minimal and maximum singular values when $m\not=d$.

Please give more concrete examples when the Jacobian can be exactly computed as required in the PHGD algorithm. If the Jacobian can be exactly computed, why one does not solve the hidden problem to optimality first and then find a mapping from control to optimal decision directly?

Missing literature: there exists some literature on hidden convex structures in optimization and reinforcement learning with non-asymptotic convergence guarantee. The hidden gradient descent also appears previously. See literature [1-2] and reference therein.

[1] Zhang, Junyu, et al. "Variational policy gradient method for reinforcement learning with general utilities." Advances in Neural Information Processing Systems 33 (2020): 4572-4583.

[2] Chen, Xin, et al. "Efficient Algorithms for Minimizing Compositions of Convex Functions and Random Functions and Its Applications in Network Revenue Management." arXiv preprint arXiv:2205.01774 (2022).

**Questions:**

P2 Line 77, the "[]" should be deleted.

P3 Line 99, Here $F$ is defined over $\theta$. $F$ is later defined over $x$. Making it super confusing. Although $F$ is represent $l$ here, later $f$ is also used to denote $l$. Another notation $L$ is used to denote the stochastic counterpart of $l$. And then $f$ is used to denote a stochastic counterpart of $L(\chi)$. Please read these notations to see how confusing they are.

P5 Line 195, it is unclear why $P$ is defined over $\theta_i$ rather than $\theta$.

P5 Lemma 1, what is a hidden smooth structure? What assumptions are exactly needed on $F$ and $\chi$ function to ensure the format of $P$.

P6, in Lemma 2, I suppose $\chi(\theta)=x$ by definition?

P6 Line 230, about the statement for the first moment is totally wrong.

P6 Line 238 & 239, $f_i$ is previously defined for deterministic function. Now $f_i$ becomes a random function.

What is exactly the algorithm that one should run? I suppose that one should run PHGD with $\hat g_{i,t}$ defined by $L$ rather than $f_i$ right?

Line 308, there should be a space between "games" and "This".

Lemma 6, the notation is broken.

Equation (C.5j) missing 1/2 in the third line.

There are many mismatch notations in the main context and the appendix.



**Limitations:**

The motivation for using the method is not adequately discussed. Some more explanation on the limitation of some assumptions is definitely welcomed.

---

> ### Author Rebuttal · Authors · 2023-08-09
>
> Thank you for your input. Before our point-by-point replies, we would only like to respectfully point out a potential misunderstanding: the fact that the latent structure may be known to the players does not imply that the problem can be solved in the latent space and the solution transferred back to the control layer. Specifically, even if $\chi$ is known (and its derivatives computable), it may still be computationally hard to invert it, so mapping latent variables to control variables is, in general, not possible. We detail this issue in our replies below.
>
>
> 1. **The [paper has] lots of typos.**
>
>     We spotted a number of typos in our original submission, which we fixed in the supplement (cf. Erratum in Appendix A). We trust that our replies below will clear up any remaining confusion.
>
>
> 2. **Example 2.1 is too simple.**
>
>     This only served as a gentle start to ease the reader into the model, hence its simple (but not simplistic) nature.
>
>
> 3. **Example 2.2, it is unclear why one does not first solve the bilinear problem and find a corresponding mapping.**
>
>     Inverting $\chi$ might be an intractable problem, as hard (or harder) than the original game. Since the aim is to find a solution in $\theta$, solving in $x$ and then inverting is not a viable approach.
>
>
> 4. **Figure 1 does not need a neural network.**
>
>     This was only meant to illustrate a standard simple example in the spirit of Mladenovic et al. We will add a pointer to clarify.
>
>
> 5. **I wonder if $\Theta$ [and $\mathcal{X}$] is the whole space. [...] Please specify the conditions for [the singular values of] the Jacobian.**
>
>    The simplify the presentation, we assumed that $\Theta = \mathbb{R}^{m}$ from the second line of Section 2. By contrast, the latent space $\mathcal{X}$ **need not be** all of $\mathbb{R}^d$ (and, indeed, in many cases of interest, it isn't).
>
>    For the singular value requirement for $\chi$, this is a mild topological condition: for example, if $\chi$ is a linear map, this simply posits that it has full column rank. The general case is the nonlinear version of this requirement, namely that $\chi$ can be uniformly approximated at each point by a full-rank linear map.
>
>
> 6. **If the Jacobian can be computed, why not solve the hidden problem and then find a mapping from control to optimal decision?**
>
>    Even though the Jacobian can be computed exactly, this does not mean that $\chi$ can be inverted. As a toy example, let $\chi(\theta_1,\theta_2) = \log\left[(1+\theta_2^2)e^{\theta_1} + (1+\theta_1^2)e^{\theta_2}\right]$: the Jacobian is trivial to calculate, but the inversion $\chi(\theta_1,\theta_2) = x$ can be very hard to compute. The difficulty of inverting $\chi$ scales exponentially in $m$ and $d$ so, in general, backsolving is not a feasible approach.
>
>
> 7. **[Missing literature]**
>
>     Thanks for bringing these papers to our attention. We will certainly discuss them, but we should also stress that our primary focus and results are notably distinct from these references.
>
>     On [1]: hidden games can indeed serve as a model for expressive neural nets. However, Assumption 4.1 of [1] stipulates the invertibility of the policy's hidden map, an assumption that we explicitly avoid. [Inverting a function represented by a neural net can be computationally prohibitive, whereas the Jacobian is typically available in closed form).
>
>    On [2]: Assumption 2.1(b) in [2] calls for the hidden map to be non-decreasing across segments of hidden parameters. This is an inherently one-dimensional consideration, while our model has been explicitly designed to treat high-dimensional problems.
>
>
> 8. **L99: $F$ is defined over $\theta$ [and later] over $x$.**
>
>     Please disregard L99, this was a typo; see L144 for the proper definition as the pseudo-gradient of $f$. As for $\ell$ and $f$, since we introduce stochasticity after defining them, randomness was implied from the context. We will update notation to make this clear.
>
>
> 9. **L195, it is unclear why $P$ is defined over $\theta_i$ rather than $\theta$.**
>
>     $\mathbf{P}_i$ should involve only the control variables of player $i$; otherwise, each player would have to know everyone else's control variables.
>
>
> 10. **Lemma 1: what is a hidden smooth structure? What assumptions are needed on $F$ and $\chi$?**
>
>     Apologies, "smooth" should read "monotone". Other than that, Lemma 1 was only stated as a stepping stone to achieve the desideratum right above and to derive the form of $P$.
>
>
> 11. **In Lemma 2, I suppose $\chi(\theta) = x$?**
>
>     Yes - cf. Fig. 1, the "Notation" paragraph at the end of Section 2, the beginning of Section 3, etc.
>
>
> 12. **Line 230, about the statement for the first moment is totally wrong.**
>
>     Yes, this statement was incorrect; please see the Errata in Appendix A.
>
>
> 13. **Lines 238 & 239, $f_i$ is previously deterministic. Now $f_i$ becomes random.**
>
>     To lighten the notation, we treated $f$ as a variadic function, taking an extra stochastic argument when needed. We did this to minimize notational clutter, but we realize the confusion, so we will correct this in the revision.
>
> 14. **One should run PHGD with $\hat g_{i, t}$ defined by $L$ rather than $f_i$ right?**
>
>     Yes, the second equality in L239 was only intended as a reminder.
>
>
> 15. **Lemma 6, the notation is broken.**
>
>     This was fixed in the Erratum (App. A).
>
>
> 16. **The motivation for using the method is not adequately discussed.**
>
>     As we explained above, the motivation is straightforward: knowledge of the hidden structure map $\chi$ does not mean that it is possible to invert it. PHGD retains all the good convergence properties one could expect in the latent, monotone space, without needing to solve a computationally prohibitive nonlinear system.
>
>
> 17. **{Minor typos}**
>
>    Thank you. We will fix them all in the revision.
>
>
> Thanks again for your input and please let us know if any of the above is not clear.

---

> > ### Comment · Reviewer_cb9a · 2023-08-10
> >
> > Thank you for the response. Despite the readability issue caused by typos, I appreciate the technical novelty in the paper.
> >
> > With regard to point 16, it seems that the proposed method is most useful when one cannot control the decisions made in the latent monotone games. Otherwise, there is no point in learning from the control layer, as one can directly play a monotone game in the hidden latent space. In this regard, the examples provided do not support the motivation well enough.

---

> > > ### Author Response · Authors · 2023-08-11
> > >
> > > First, we would like to sincerely thank you for your quick response and input.
> > >
> > > The first thing we would like to clarify is that, in line with the established literature on hidden games (see below for a representative list of references), the agents *do not play the hidden game directly*: the agents' decisions are their control variables, not the latent variables. The latent variables merely represent a modeling abstraction and are best thought of as virtual/auxiliary variables that capture a certain structure in the players' payoff functions. In this regard, identifying a solution in terms of latent variables is not meaningful from the players' viewpoint unless they can also find the control/decision variables that realize said solution.
> > >
> > > We should also note that this model is not due to our paper but is the standard model for hidden games in the literature, see references [1-4] below. Given that this is an established model in the literature, the main focus of our paper was to develop a suite of learning algorithms and tools that would allow the effective solution of such problems.
> > >
> > > As for additional instances/examples of our model, another interesting class of problems that can be cast in the framework of hidden games is that of team zero-sum games [5], where there exist two competing teams of agents. For each possible outcome of the game, the payoffs of all members within a single team are equal to each other and represent the payoff of the team. The sum of the payoffs of two teams is equal to zero. Several variations of these models have been the object of recent study [6-8] and they readily fall within a hidden game framework where the space of latent variables is the set of (mixed extensions) of the strategy outcomes of each team.
> > >
> > > We hope that the above clarifies further the positioning of our work in the existing literature on hidden games.
> > >
> > > We thank you again for your input and please let us know if you have any further questions.
> > >
> > > > [1] Vlatakis et al. Poincare recurrence, cycles and spurious equilibria in gradient-descent-ascent for non-convex non-concave zero-sum games. (NeurIPS 2019)
> > >
> > > > [2] Flokas et al. Solving min-max optimization with hidden structure via gradient descent ascent. (NeurIPS 2021)
> > >
> > > > [3] Mladenovic et al. Generalized natural gradient flows in hidden convex-concave games and gans. (ICLR 2021)
> > >
> > > > [4] Pattathil et al. Symmetric (optimistic) natural policy gradient for multi-agent learning with parameter convergence. (AISTATS 2023)
> > >
> > > > [5] Schulman et al. The duality gap for two-team zero-sum games. Games and Economic Behavior (2019)
> > >
> > > > [6] Anagnostides et al. Algorithms and Complexity for Computing Nash Equilibria in Adversarial Team Games. (EC 2023).
> > >
> > > > [7] Kalogiannis et al. Teamwork makes von Neumann work: Min-max optimization in two-team zero-sum games. (ICLR 2023)
> > >
> > > > [8] Kalogiannis et al. Efficiently Computing Nash Equilibria in Adversarial Team Markov Games. (ICLR 2023)

---

> > > > ### Comment · Reviewer_cb9a · 2023-08-14
> > > >
> > > > Thanks for the clarification. I have read all responses and will raise the score to 5. I believe serious efforts need to be made in order to improve the presentation quality and correct all typos but the merit of the paper is also clear.

---

> > > > > ### Author Response · Authors · 2023-08-18
> > > > >
> > > > > Thank you for your continued input and for upgrading your score, we will update our manuscript according to your detailed suggestions and remarks.
> > > > >
> > > > > Regards,
> > > > >
> > > > > The authors

---

### Author Rebuttal · Authors · 2023-08-10

Dear AC, dear reviewers,

We are sincerely grateful for your time and constructive input. To streamline the discussion phase, we reply to each reviewer’s questions in a separate point-by-point thread below.

Kind regards,

The authors

---

### Decision · Program_Chairs · 2023-09-21

**Decision:**

Accept (poster)

**Comment:**

The paper makes solid contribution in introducing a preconditioned hidden gradient descent algorithm for solving a general class of hidden monotone games. Theoretical results are novel, clearly presented, and supported by several non-trivial numerical experiments.


The authors' response has carefully addressed most concerns brought up by the reviewers in light of the assumptions and comparison to existing work. There are some lingering concerns on the presentation quality. I think the contribution outweighs the weakness and therefore would recommend to accept the paper. In the final version, please follow the reviewers' suggestion and significantly improve the presentation of the paper as promised.